# Altered milk tryptophan and tryptophan metabolites in women living with HIV

Nicole H. Tobin [1], Fan Li[1], Wentao Zhu[2], Kathie G. Ferbas[1], John W. Sleasman[3], Daniel Raftery [2], Louise Kuhn[4] & Grace M. Aldrovandi[1] ✉

Children born to women living with HIV (WLWH) suffer increased morbidity and, in low-income settings, have two to three times the mortality of infants born to women without HIV. The basis for this increase remains elusive. In low-income settings, breastfeeding is recommended because health benefits outweigh the risk of transmission, especially when maternal antiretroviral therapy is provided. We profile the milk metabolome of 326 women with and without HIV sampled longitudinally for 18 months postpartum using global metabolomics. We identify perturbations in several metabolites, including tryptophan, dimethylarginine, and a recently discovered antiviral ribonucleotide, that are robustly associated with maternal HIV infection. Quantitative tryptophan and kynurenine levels in both milk and plasma reveal that these perturbations reflect systemic depletion of tryptophan and alterations in tryptophan catabolism in WLWH. Finally, we validate these signatures of maternal HIV infection in an independent cohort of healthier WLWH. Taken together, our findings demonstrate that milk tryptophan content and availability decrease among WLWH, which may indicate perturbations in milk tryptophan catabolism. The link between this perturbation and the increased morbidity and mortality of children born to WLWH merits further investigation.

Children born to women living with HIV (WLWH) experience significantly increased morbidity and mortality, even when they themselves are not infected (HIV-exposed, or CHEU). While CHEU in high-income countries are more likely to be hospitalized[1–3], those in low-income countries have mortality rates two to three times higher than infants born to women without HIV (WWoH)[4–7]. Most of this morbidity and mortality is due to infections, particularly pneumonia, diarrhea, and meningitis[7,8]. Additionally, CHEU faces challenges related to growth and cognitive development[7]. Although maternal antiretroviral therapy has improved CHEU health outcomes, they still experience significant risks, the basis of which remains unclear.

HIV infection, even with optimal viral suppression by antiretroviral therapy, is associated with chronic inflammation characterized by persistent macrophage activation, elevated levels of interferon-inducible cytokines, and intestinal barrier dysfunction[9]. This systemic inflammation also affects pregnant WLWH and is reflected in their CHEU infants[10–14]. Similar to their mothers, CHEU infants display perturbed biomarkers of immune monocyte activation, including elevated interferon-inducible cytokines, which impair immune function and germinal center development throughout infancy[15–18]. Such changes are not unique to HIV as maternal immune activation due to maternal influenza and other infections is associated

[1]Division of Infectious Diseases, Department of Pediatrics, David Geffen School of Medicine at the University of California, Los Angeles, CA, USA. [2]UW Northwest Metabolomics Center, Seattle, WA, USA. [3]Division of Pediatric Allergy and Immunology, Department of Pediatrics, Duke University School of Medicine, Durham, NC, USA. [4]Gertrude H. Sergievsky Center, Vagelos College of Physicians and Surgeons; and Department of Epidemiology, Mailman School of Public Health, Columbia University Irving Medical Center, New York, NY, USA. ✉e-mail: GAldrovandi@mednet.ucla.edu

with the development of neurodevelopmental disorders[19–24] as well as the development of tissue-specific immunity and inflammation[25–27] in their offspring.

HIV-associated immune dysregulation is largely driven by metabolic alterations, particularly in the indoleamine 2,3-dioxygenase (IDO) pathway and accelerated tryptophan catabolism, both of which impair immune function[28,29]. Tryptophan, an essential amino acid obtained solely from the diet, is primarily catabolized via the kynurenine pathway, with its metabolites playing key roles in immune regulation, neuronal function, and intestinal homeostasis[30–32]. A key marker of IDO activation is the kynurenine-to-tryptophan (KT) ratio[33], which is elevated in HIV infection and associated with disease progression, dementia, and mortality[34]. Although the KT ratio decreases with viral suppression, it does not normalize completely[35,36]. High KT ratios are linked to microbial dysbiosis, TH17 cell loss, and an imbalance between TH17 and Treg cells in the gastrointestinal mucosa[29,37]. Additionally, elevated KT ratios have been associated with stunting in children[38,39], suggesting that disrupted tryptophan metabolism may have broader health impacts.

Breast milk is essential for optimal growth and development, particularly for infants born to WLWH in low- and middle-income countries. The dietary source of tryptophan for nursing infants is human milk. There are very few reports describing levels of tryptophan in milk, it appears that tryptophan, free and bound, are higher in colostrum and then decline with time[40–44]. Utilizing longitudinal samples from a cohort of WLWH and WWoH prior to the routine use of antiretroviral therapy, we characterize the metabolomic profile of breast milk over the first 18 months postpartum. We identify a metabolomic signature of milk in WLWH that demonstrates decreased levels of tryptophan throughout lactation. This depletion of tryptophan is associated with elevations of the type I interferon response[28]. Using targeted assays for tryptophan and kynurenine in both milk and plasma, we demonstrate that quantitative levels of tryptophan and kynurenine are systemically altered in WLWH. Furthermore, in later experiments we identify an initially unknown compound, which is highly elevated in the milk of WLWH, as 3′-deoxy-3′4′-didehydro-cytidine (ddhC), the free base of an interferon-inducible innate antiviral ribonucleotide[45]. Lastly, we confirm this milk metabolomic signature

of HIV in a second, healthier cohort of WLWH on antiretroviral therapy from Haiti (see study overview, Fig. 1).

## Results
### More than eight-hundred metabolites characterized over 18 months of lactation in 326 women

To investigate the milk metabolome of WLWH, global metabolomics was performed on 1599 whole human milk samples collected as part of a randomized clinical trial[46]. The Zambia Exclusive Breastfeeding Study (ZEBS) was conducted in Lusaka, Zambia, between 2001 and 2008 and investigated the feeding strategy for WLWH recommended by the World Health Organization at that time. This trial was conducted prior to the routine use of antiretroviral therapy, with most women receiving single-dose nevirapine for prevention of vertical transmission per standard of care. Women/infant pairs were followed for 24 months postpartum. Longitudinal breast milk samples spanning the first 24 months of lactation were selected from 38 WWoH and a stratified sample of 288 WLWH enrolled in the trial. The clinical characteristics of the study participants are shown in Table 1. Thirty-six (12.5%) of WLWH died during the course of the study compared to no WWoH ($p < 0.05$). WLWH had lower CD4 T cell counts (315 vs. 840 cells/mm$^3$) and higher CD8 T cell counts (759 vs 563 cells/mm$^3$) than WWoH ($p < 0.001$ and $p = 0.001$, respectively). Children of WWoH weighed more at birth (3.3 vs 2.9 kg; $p < 0.01$) and 3 of these children died during the study.

All but 2 of the 1599 whole breast milk samples were successfully characterized and included 765 named and 74 unnamed metabolites for a total of 839 metabolites. Of the 765 named metabolites, 20 metabolites classified as "drug-antibiotic" or "drug-antiviral" were removed from further analyses, leaving a total of 819 metabolites included in the analyses. Samples were excluded from this analysis if there were insufficient samples at that timepoint for meaningful interpretation ($n = 72$) or if the milk was collected post-weaning ($n = 99$), leaving 1,426 samples spanning 18 months of lactation in the final analysis (Fig. 1A and Supplementary Fig. 1).

### Metabolomic signature of maternal HIV infection in milk

Overall milk metabolomic profiles were strongly influenced by study visit (4.0% variance, $p < 0.001$), and by HIV infection (0.19% variance,

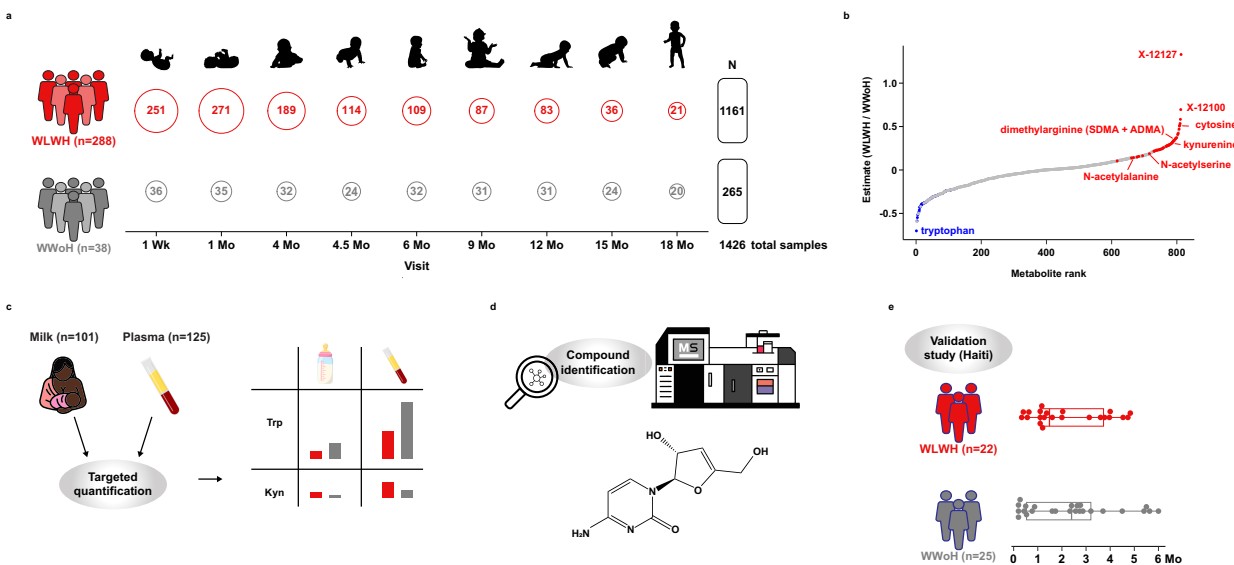

**Fig. 1 | Study Design. a** Large, global metabolomic study of milk of WLWH versus WWoH. Open circles denote the number of samples at each study visit included in the final analysis. The number of participants in each group is indicated in parentheses on the left, and the number of samples is indicated on the right.
**b** Differentially abundant metabolites in the milk WLWH vs WWoH, shown in rank order. **c** Targeted quantification of tryptophan and kynurenine levels in maternal milk and plasma. **d** Identification of the unknown compound X-12127 as 3′-deoxy-3′,4′-didehydro-cytidine (ddhC). **e** Validation of milk metabolomic findings in an independent cohort of WLWH from the Haiti study. Points show the timing of sample collections, and boxplots indicate the median and interquartile ranges.

## Table 1 | Baseline Characteristics of Study Cohorts

| Baseline Characteristics of Zambia Exclusive Breastfeeding Study Cohort | | | |
|---|---|---|---|
| Value (mean (Standard Deviation)) | WWoH | WLWH | *p*-value |
| N | 38 | 288 | |
| Maternal Age (years), mean (SD) | 26 (7) | 26 (5) | 0.542 |
| Maternal BMI, mean (SD) | 22.3 (4.1) | 21.3 (3.0) | 0.061 |
| Parity, mean (SD) | 2.3 (1.9) | 2.5 (1.7) | 0.510 |
| Maternal education (years, mean (SD)) | 7.6 (3.0) | 7.2 (2.9) | 0.491 |
| Food unavailable within the past month (n (%)) | 11 (28.9) | 79 (27.4) | 0.997 |
| Household electricity available (n (%)) | 25 (65.8) | 178 (61.8) | 0.766 |
| Log10 HIV RNA, mean (SD) | NA | 4.81 (0.76) | |
| Maternal CD4 Count (cells/mm$^3$) | | | |
| Mean (SD) | 840 (246) | 315 (185) | <0.001 |
| <200 cells/mm$^3$ (n (%)) | 0 (0.0) | 94 (32.6) | |
| 200–349 cells/mm$^3$ (n (%)) | 2 (5.3) | 98 (34.0) | |
| >350 cells/mm$^3$ (n (%)) | 36 (94.7) | 96 (33.3) | |
| Maternal CD8 Count (cells/mm$^3$), mean (SD) | 563 (233) | 759 (345) | 0.001 |
| Maternal CD3 Count (cells/mm$^3$), mean (SD) | 1481 (434) | 1131 (450) | <0.001 |
| Cesarean Section (%) | 1 (2.6) | 5 (1.7) | 1 |
| Nevirapine at Delivery (n (%)) | NA | 277 (96.2) | NA |
| Gestational Age at Delivery (weeks), mean (SD) | 38.8 (5.4) | 37.8 (4.1) | 0.186 |
| Infant Birth Weight (kg), mean (SD) | 3.25 (1.24) | 2.94 (0.49) | 0.005 |
| Infant Male Sex (n (%)) | 13 (34.2) | 143 (49.7) | 0.106 |
| 12-month CD4 Count (cells/mm$^3$), mean (SD) | 927 (349) | 369 (298) | <0.001 |
| Child death (%) | 3 (7.9) | 145 (50.3)* | NA |
| Maternal death (%) | 0 (0.0) | 36 (12.5) | 0.042 |
| **Baseline Characteristics of Haiti Cohort** | | | |
| Value (median (Standard Deviation)) | WWoH | WLWH | p-value |
| N | 25 | 22 | |
| Maternal Age (years), mean (SD) | 27 (7) | 30 (6) | 0.119 |
| Maternal BMI, mean (SD) | 22.0 (2.8) | 24.1 (3.8) | 0.028 |
| Parity | 3.0 (1.9) | 3.3 (1.8) | 0.565 |
| Maternal educational attainment (n (%)) | | | 0.003 |
| No school | 5 (20.0) | 5 (22.7) | |
| Basic education | 14 (56.0) | 5 (22.7) | |
| Lower secondary | 6 (24.0) | 3 (13.6) | |
| Upper secondary | 0 (0.0) | 9 (40.9) | |
| Maternal mid-upper arm circumference (cm), mean (SD) | 26.0 (4.1) | 26.4 (4.3) | 0.75 |
| Log10 HIV RNA, mean (SD) | | 1.59 (1.45) | NA |
| Plasma HIV RNA (copies/mL) | | | NA |
| <1000 (n (%)) | | 15 (68.2) | |
| >1000 (n (%)) | | 7 (31.8) | |
| Maternal CD4 Count (cells/mm$^3$) | | | NA |
| Mean (SD) | NA | 554 (271) | |
| <200 cells/mm$^3$ (n (%)) | | 1 (4.5) | |
| 200–349 cells/mm$^3$ (n (%)) | | 3 (13.6) | |
| >350 cells/mm$^3$ (n (%)) | | 18 (81.8) | |
| Maternal CD4 Count (cells/mm$^3$) | | | NA |
| <350 cells/mm$^3$ (n (%)) | | 4 (18.2) | |
| >350 cells/mm$^3$ (n (%)) | | 18 (81.8) | |
| Cesarean Section (%) | 1 (4.0) | 5 (22.7) | 0.138 |
| Infant Age in Days, mean (SD) | 71 (57) | 66 (46) | 0.725 |
| * Based on sampling strategy | | | |

$p < 0.001$), maternal CD4 count (0.14% variance, $p < 0.001$), maternal BMI (0.12% variance, $p < 0.001$), and infant sex (0.07% variance, $p = 0.009$) (Supplementary Fig. 2 and Supplementary Table 1). Linear mixed effects regression identified 103 metabolites with significantly altered levels in WLWH versus WWoH when all study visits were considered in aggregate (Supplementary Data 1). Post hoc comparisons stratified by study visit identified a core set of altered metabolites (Fig. 2A and Supplementary Fig. 3; Supplementary Data 2), including tryptophan, dimethylarginine, cytosine, and an uncharacterized compound, X-12127. Levels of tryptophan, dimethylarginine, cytosine, and X-12127 were significantly and consistently altered in the milk of WLWH across the entire study time course ($p < 0.001$; Fig. 2B, red stars).

As expected, tryptophan, cytosine and X-12127 were strongly correlated with baseline maternal CD4 count, baseline maternal plasma viral load, and contemporaneous breast milk viral load (Supplementary Fig. 4 and Supplementary Data 3). N-acetylalanine ($p = 0.006$) and N-acetylserine ($p = 0.007$) were elevated in WLWH with greater separation at the earlier timepoints, 1 and 4 months. In contrast, the linoleic acid metabolites 9,10-DiHOME and 12,13-DiHOME were significantly elevated in WLWH at 1 month ($p < 0.05$), but not over the entire study time course. X-07765, an unknown metabolite significantly elevated at the one-month timepoint in the linear regression model ($p = 0.003$), did not appear to be as robust a marker as the other metabolites and was not significant over the entire time course ($p = NS$; Fig. 2B). Other features, including many lipid and amino acid metabolites, were found to be altered later in lactation (Supplementary Fig. 3 and Supplementary Data 4) and may reflect the far-reaching impact of HIV infection on milk composition. In an orthogonal analysis, random forests models for maternal HIV status also identified many of these same features as significant predictors of maternal HIV (Fig. 2B black stars, Supplementary Fig. 5, Supplementary Data 5). Finally, sub-analyses of WLWH whose children remained uninfected during the course of the study (38 WWoH versus 154 WLWH with CHEU) and those whose children survived through the entire study course (35 WWoH versus 143 WLWH) confirmed the metabolic signature of maternal HIV infection to be independent of child outcome (Supplementary Fig. 6 and Supplementary Data 6–9). Taken together, these results point to a robust and sustained effect of HIV infection on the composition of metabolically relevant compounds in human milk.

### Alteration in tryptophan and its metabolites in the milk of WLWH

The striking separation in normalized abundances of tryptophan and X-12127 across the study course (Fig. 2B) led to further investigations of these two metabolites. Of note, although tryptophan abundance declined slightly between the 1-week and 1-month milk samples in WWoH, there was a notable dip in tryptophan abundance in the WLWH, followed by some recovery and continued significant separation of tryptophan throughout the study course. The normalized abundance of X-12127 rose sharply between 1 week and 1 month and remained elevated, a pattern that was mirrored by cytosine. It was also noted that dimethylarginine, N-acetylalanine, and N-acetylserine all had similarly shaped normalized abundance curves.

The significant depletion of tryptophan across the study duration was mirrored by an increase in levels of X-12100 (tentatively identified as a hydroxylated tryptophan compound) as well as a consistent, albeit smaller, increase in levels of kynurenine (Supplementary Fig. 7 and Supplementary Data 1-2). Although none of the comparisons in the analysis stratified by visit demonstrated significant differences after adjustment for multiple comparisons (Supplementary Data 2), kynurenine was increased in the mean effects model ($p = 0.007$, Supplementary Data 1). Moreover, the kynurenine to tryptophan (KT) ratio, a marker of IDO activity, was significantly elevated at all study visits (Fig. 3A and Supplementary Table 2; $p < 0.001$).

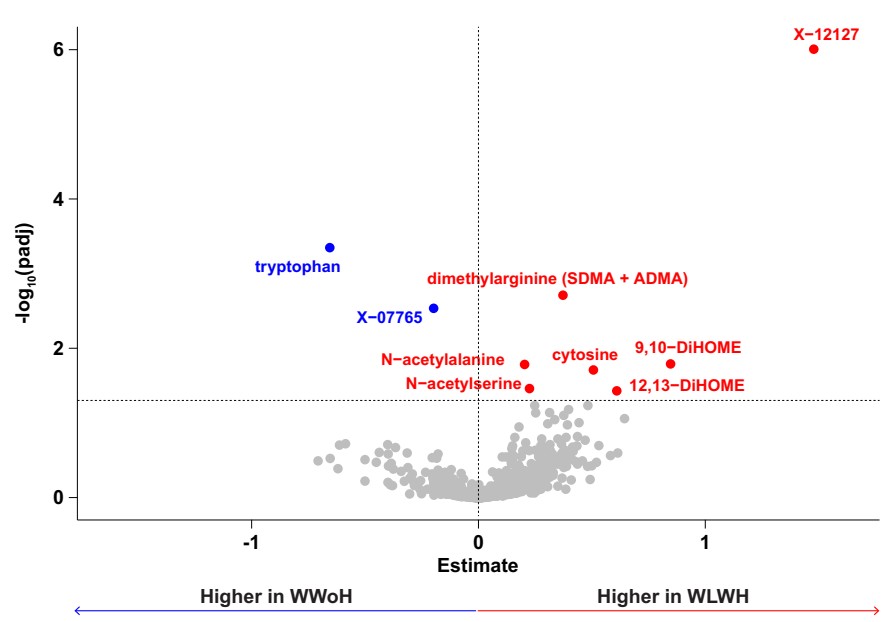

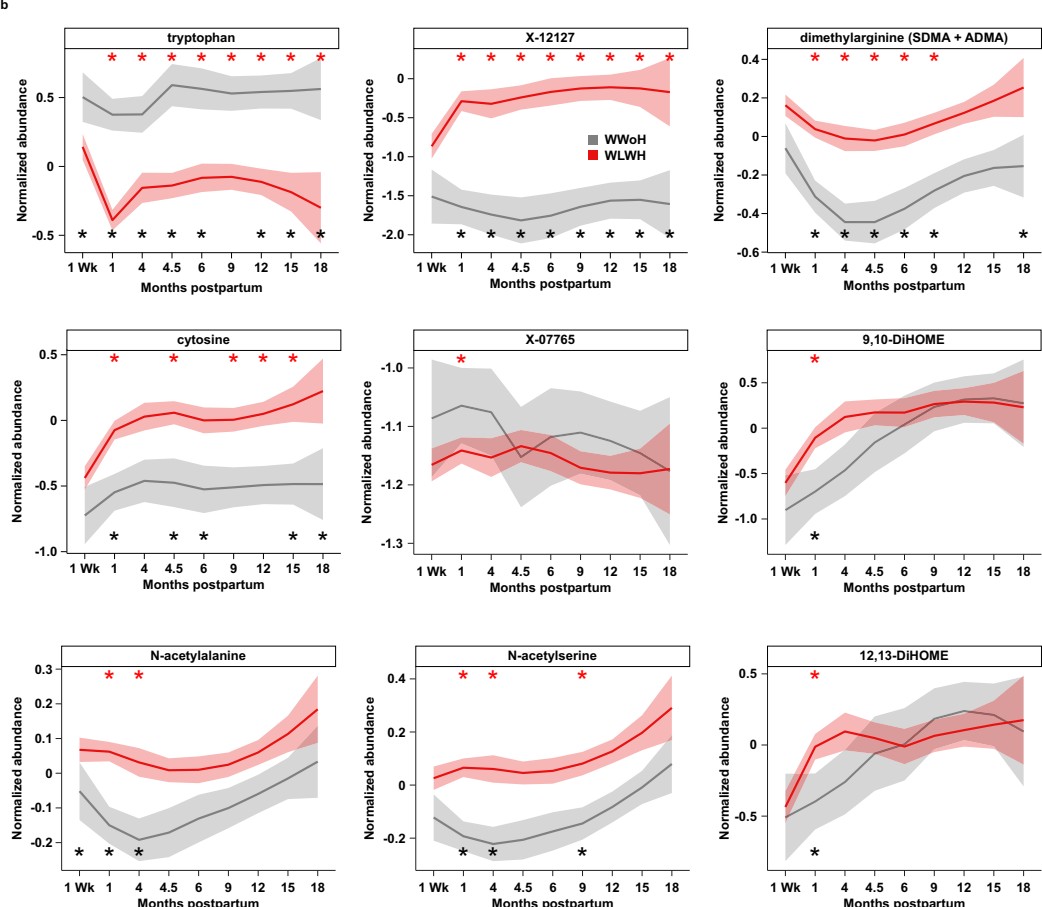

**Fig. 2 | Metabolomic signature of HIV infection in milk. a** Volcano plot showing differences in metabolite abundances at the 1-month timepoint. Blue and red points metabolites are significantly decreased and increased in WLWH versus WWoH, respectively. **b** Normalized abundances of selected metabolites across the study course. Solid lines indicate mean abundances, and shaded areas denote 95% confidence intervals. Red asterisks along the top denote study visits at which the selected compound was differentially abundant in WLWH versus WWoH. Black asterisks along the bottom denote study visits at which the compound was selected as a predictive feature in the random forests modeling. All metabolites that were significantly differentially abundant at the 1-month timepoint are shown.

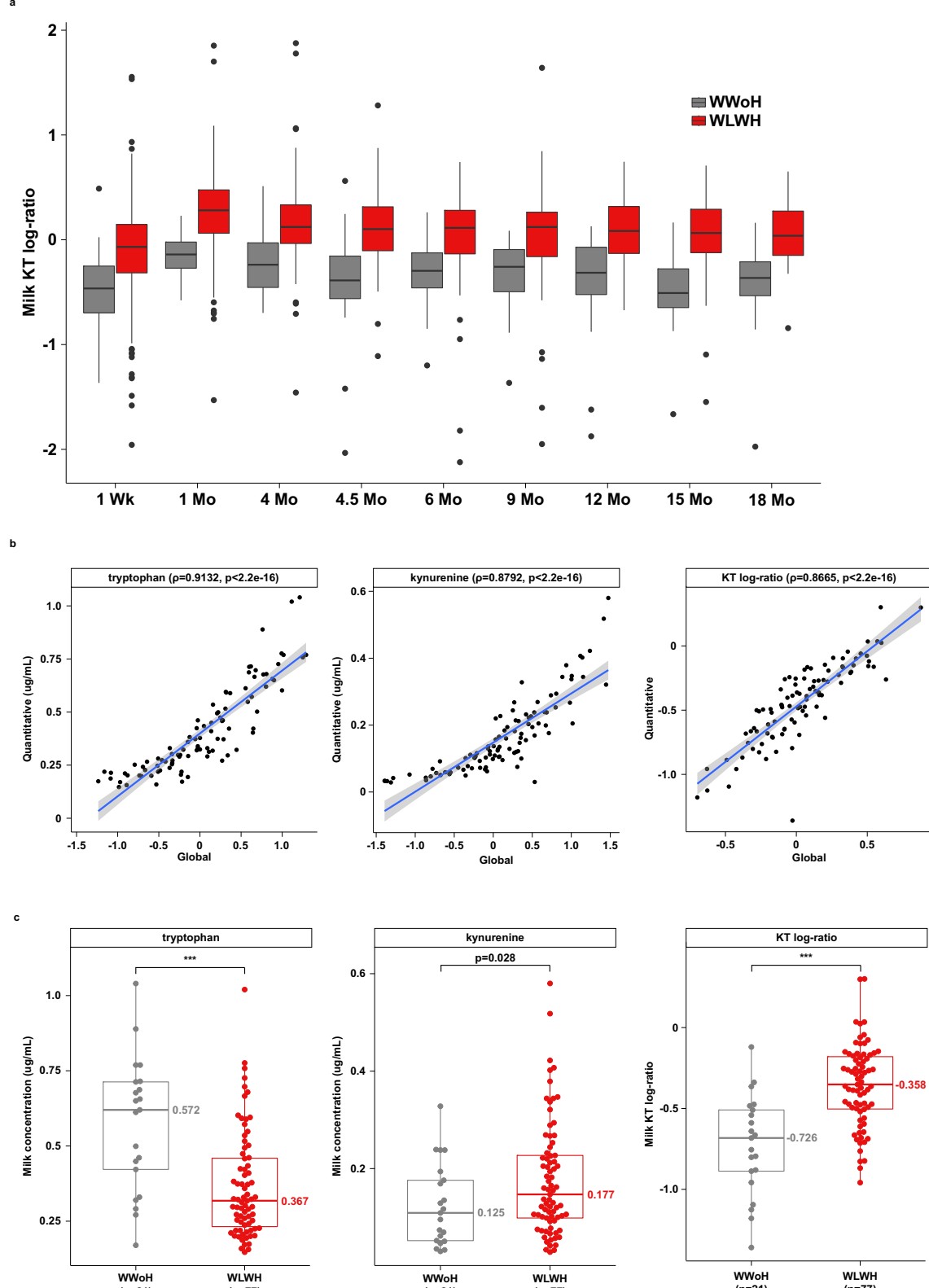

**Fig. 3 | Altered tryptophan and kynurenine levels in the milk of WLWH.**
**a** Boxplot of kynurenine/tryptophan log-ratio in WLWH versus WWoH stratified by study visit, as calculated from the global metabolomics panel ($n = 1426$; linear regression adjusted $p < 0.001$ all timepoints). **b** Scatterplots of global versus quantitative tryptophan and kynurenine abundances and KT log-ratio at the 4-month timepoint ($n = 98$). Solid blue line indicates linear regression line and shaded gray area represents 95% confidence intervals. Spearman correlation coefficients and p-values are shown in parentheses in each subtitle. **c** Tryptophan, kynurenine, and KT log-ratio values in WLWH versus WWoH at the 4-month study visit, as calculated from the quantitative panel. Two-sided Welch's t-test * $p < 0.05$, ** $p < 0.01$, *** $p < 0.001$. Box plots show the center line as the median, whiskers show the maxima and minima, and box limits show interquartile ranges. WLWH: Women living with HIV; WWoH: Women without HIV.

To confirm the findings of the elevated KT ratio from the global metabolomics panel, a quantitative KT panel was performed on 98 milk samples from 21 WWoH and 77 WLWH at the 4-month timepoint (Supplementary Table 3). The 4-month timepoint was selected because paired milk and plasma samples were available from almost all participants at this timepoint due to the trial design. Quantitative and global KT values were highly correlated (Fig. 3B, rho = 0.9132 and 0.8792 for tryptophan and kynurenine, respectively; both $p < 2.2e{-}16$), indicating that the global panel could be used for accurate estimation of the quantitative KT-ratio. Milk tryptophan levels were significantly lower, 0.367 ug/mL versus 0.572 ug/mL ($p < 0.001$), and kynurenine levels were significantly higher, 0.177 ug/mL versus 0.125 ug/mL ($p = 0.028$) in WLWH versus WWoH (Fig. 3C and Supplementary Table 4). Correspondingly, the KT log-ratio was significantly higher in WLWH as well ($p < 0.001$, Fig. 3C and Supplementary Table 4).

### Lower milk tryptophan levels mirror lower plasma tryptophan levels

The lower levels of tryptophan in the milk of WLWH could reflect lower plasma levels of tryptophan in WLWH, decreased transfer of tryptophan to the milk of WLWH, or altered tryptophan metabolism in the milk of WLWH. Given that tryptophan transport across the gut can be perturbed by infection[47], we sought to determine whether transport across the breast epithelium was similarly affected. To distinguish between these possibilities, quantitative plasma and milk tryptophan and kynurenine levels were compared from 118 women (31 WWoH and 87 WLWH, Supplementary Fig. 8 and Supplementary Table 3). As expected, milk and plasma levels were highly correlated (rho = 0.50, $p < 0.001$ for kynurenine; rho = 0.66, $p < 0.001$ for tryptophan). Plasma tryptophan levels were significantly lower, 5.03 ug/mL versus 8.13 ug/mL ($p < 0.001$), and kynurenine levels were significantly higher, 0.468 ug/mL versus 0.369 ug/mL ($p < 0.001$) in WLWH versus WWoH (Supplementary Fig. 9 and Supplementary Table 4 and Supplementary Data 10).

To further investigate the transfer of tryptophan or kynurenine to milk, we compared the quantitative levels in 92 women (21 WWoH and 71 WLWH, Supplementary Fig. 8 and Supplementary Table 3) who had both plasma and breast milk samples collected at the 4-month timepoint (Supplementary Fig. 10A). Milk levels of tryptophan were 6.8% and 8.3% of the plasma levels for WWoH and WLWH, respectively; this proportion did not differ significantly by maternal HIV infection ($p = 0.14$, Supplementary Fig. 10B and Supplementary Table 4). A similar trend was observed for kynurenine with breast milk levels at 32.4% and 36.6% of the corresponding plasma levels for WWoH and WLWH, respectively ($p = 0.42$, Supplementary Fig. 10B and Supplementary Table 4). Altogether, these results suggest that the decreased levels of tryptophan and increased levels of kynurenine observed in the milk of WLWH reflect plasma levels as opposed to selective transfer to the milk or altered local metabolism.

### Identification of ddhC, a free base of the naturally occurring antiviral ribonucleotide, ddhCTP

Levels of an unknown compound, initially labeled X-12127, were significantly elevated in WLWH compared to WWoH at all study visits except immediately postpartum (Fig. 2B and Supplementary Data 2). Given the robustness of this marker, we sought to better characterize this compound by mass spectrometry. The compound, with an m/z value of 226.0824 in positive ionization mode, was analyzed using tandem MS/MS, revealing a cytosine fragment and two water loss events, suggesting the presence of two OH groups and a composition similar to 2′-deoxycytidine (dC) minus two hydrogen atoms (Fig. 4). Structural similarity exploration and predicted fragmentation indicated 3′-deoxy-3′,4′-didehydro-cytidine (ddhC) as a likely candidate. This was confirmed through standard acquisition and extensive comparison of fragmentation patterns at different collision energies,

leading to the definitive identification of the compound as ddhC (Fig. 4E).

### Validation of milk metabolomic signature of HIV in a healthier cohort of WLWH

To assess the generalizability of these findings, global metabolomics profiling was performed on the milk from a previously described cohort of 22 WLWH and 25 WWoH from Haiti (Table 1)[48]. Notably, these women were on antiretroviral therapy with a mean CD4 count of 541 cells/mm$^3$, and milk was sampled at a single visit during the first 6 months of lactation (median 45 days; Fig. 1E). Regression estimates of the associations between HIV status and metabolites were obtained while including infant age in days as a covariate given the non-uniformity of sampling. Nevertheless, estimates from the Haiti cohort were strongly correlated to those from the main cohort at the 1 month and 4 month visits (r = 0.126 and 0.386, $p < 0.001$ for all metabolites, respectively; r = 0.812 and 0.789, $p < 0.001$ when only the metabolites observed to be significantly associated with HIV status in the ZEBS cohort were included; Supplementary Fig. 11A). Three metabolites were significantly increased in the milk of WLWH, cytosine ($p < 0.001$), methionine sulfone ($p = 0.090$) and a different unknown metabolite, X-12193 ($p = 0.054$) (Supplementary Fig. 11B and Supplementary Data 11). Tryptophan trended lower and kynurenine trended higher in the milk of WLWH in the Haiti cohort as observed in the main cohort, although neither reached statistical significance. Notably, the KT log-ratio was significantly higher in the milk of WLWH (unadjusted p = 0.05).

## Discussion

Approximately 1.3 million children are born to WLWH annually, and despite maternal antiretroviral therapy, these children continue to experience differences in immune function, growth, and cognition. In this study, we find that the milk of WLWH is characterized by decreased levels of tryptophan and perturbations of tryptophan catabolism via the kynurenine pathway. These interferon-inducible metabolic alterations may serve as a common denominator to explain some of the increased morbidity and mortality experienced by CHEU.

Normal immune function depends on the regulation of key components of innate and adaptive immunity in lymphoid tissues. Interplay between antigen-presenting cells, such as macrophages and dendritic cells, is critical to host defense by maintaining mucosal barriers and attenuating inflammation. Immune dysregulation mediated by cellular immunity is associated with alterations in indoleamine 2,3-dioxygenase and tryptophan catabolism in individuals with HIV[29]. Chronic viral infections are also known to alter tryptophan catabolism[31]. Recent studies of post-acute sequelae of SARS-CoV-2 infection (PASC) demonstrate that individuals with long COVID have decreased circulating serotonin levels secondary to viral RNA-induced type I interferons, reducing tryptophan uptake through the gut epithelium[47]. Additionally, a tryptophan-deficient diet is sufficient to lead to circulating tryptophan and serotonin deficiency in a mouse model[47]. Tryptophan starvation appears to be part of the innate immune response to bacterial and parasitic infections[49]. Tryptophan catabolism, much of which is carried out by intestinal microbes, serves an essential role in physiological homeostasis and immune regulation and can rapidly be adapted in the stress response[32]. HIV infection may stimulate IDO-1 activity through virally associated increases in type I and II interferons and microbial products. Multiple tryptophan metabolites are endogenous ligands of the aryl hydrocarbon receptor (AHR), which serves as an environmental sensor that integrates immune responses[50]. Thus, deficiency or perturbation of tryptophan catabolites can lead to alterations of the gut microbiota and immunity by alterations in the IDO1-AHR axis[51]. Importantly, T helper cell subsets, including T$_H$17 and

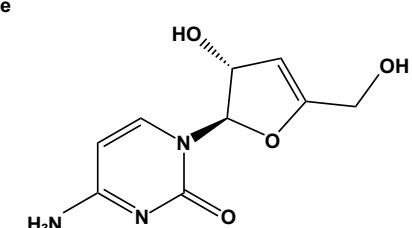

**Fig. 4 | Identification of unknown compound X-12127 as 3'-deoxy-3',4'-didehydro-cytidine (ddhC). a** Comparison of retention time (RT) for X-12127 and ddhC standard. MS/MS spectra of X-12127 and ddhC standard obtained on an Agilent 6546 LC/Q-TOF MS system in the positive targeted MS/MS scan mode by varying the collision energy (CE) at 5 ev (**b**), 20 ev (**c**), and 40 ev (**d**). **e** Chemical structure of 3'-deoxy-3',4'-didehydro-cytidine (ddhC).

regulatory T cells, express AHR, so alterations in these pathways may alter immune homeostasis and immune development in children of women living with HIV[50]. In the context of growth, tryptophan serves as a limiting amino acid for protein metabolism. Additionally, alterations in tryptophan catabolism have been associated with stunting[38,39]. Cognitive effects of tryptophan depletion are thought to be due in part to decreased availability of serotonin, as serotonin is critical for neuronal differentiation, migration, and synapse formation during embryonic development[51]. Altered tryptophan catabolism may also explain some of the neurocognitive deficits in children of WLWH by leading to increases in neurotoxic metabolites, such as quinolinic acid, and decreases in neuroprotective metabolites, such as kynurenic acid[30]. Kynurenine and quinolinic acid are elevated in the cerebrospinal fluid of people with HIV and correlate with the severity of HIV-associated neurocognitive disorder[31,52].

Type I interferons also lead to the production of 3'-deoxy-3',4'-didehydro-cytidine triphosphate (ddhCTP), an innate antiviral compound produced by the interferon-inducible protein *viperin* (*vir*us *i*nhibitory *p*rotein, *e*ndoplasmic *r*eticulum-associated, *in*terferon-inducible)[45]. ddhC, which is highly elevated in the milk of WLWH, is the free base of this naturally occurring antiviral ribonucleotide and is also being investigated as a biomarker for acute viral infection[53,54]. ddhCTP has direct antiviral properties and acts as a chain terminator for the RNA-dependent RNA polymerases from multiple members of the Flavivirus genus[45], Zika Virus, and likely SARS-CoV-2[54–56]. Furthermore, HIV-1 infection of macrophages induces *viperin*-mediated inhibition of viral replication[57]. *Viperin* is a multifunctional protein and product of an ancient gene that is part of the radical *S*-adenosyl-L-methionine (SAM) superfamily of enzymes[56]. The metabolic pathways, products, and serum cytokine associations of *viperin* are being elucidated[54]. Viperin is co-transcribed with cytidylate monophosphate kinase 2 (*CMPK2*) during interferon stimulation. It has been proposed that CMPK2 primarily functions to produce enough substrate for the viperin-mediated production of ddhCTP, which catalyzes the conversion of cytidine triphosphate to ddhCTP.

Cytosine was also persistently elevated in the milk of WLWH. Viral infection has been associated with alterations in host cytosine metabolism[58] and the production of ddhCTP[59]. While ddhCTP is a viral chain terminator, host RNA and DNA polymerases are not affected by ddhCTP[45]. We found cytosine elevations in both our primary cohort and our validation cohort. We hypothesize that the elevated levels of cytosine in the milk of WLWH reflect the breakdown of ddhCTP. The elevation of dimethylarginine alone and with the N-acetylated amino acids (N-acetylalanine and N-acetylserine) in our cohort is consistent with oxidative stress and increased protein catabolism. Our study does not discriminate between symmetric versus asymmetric dimethylarginine. There is emerging data on symmetric arginine methyltransferases potentially playing an important role in immune cell development and homeostasis[60]. Alterations in T cell development may underpin the immunologic abnormalities seen in children of WLWH.

These findings raise the question of whether replacement of tryptophan in the diet of children of WLWH, if low tryptophan is found in investigations of CHEU, will lead to resolution of the immune, cognitive, and growth perturbations experienced by this population. It is not known if the low milk tryptophan levels are sufficient, how well the tryptophan is absorbed, or the proportion of bound tryptophan in the milk. Systemic inflammation has been described in CHEU infants, which may alter tryptophan catabolism, leading to increases in neurotoxic metabolites. These possibilities deserve careful study. Breastfeeding remains the recommended form of infant feeding because health benefits outweigh the risk of HIV transmission. With effective maternal antiretroviral therapy, HIV transmission risk has decreased to the point where shared decision-making about breastfeeding entered US guidelines in 2023. The

decreased levels of milk tryptophan in the setting of chronic HIV infection are likely due to downregulation of tryptophan transporters in the maternal gut, resulting in lower tryptophan plasma levels[47]. Whether the pro-inflammatory milieu of milk similarly affects infant tryptophan absorption is unknown. Prudence is also demanded as inflammatory imprinting from the milk of WLWH might drive tryptophan catabolism down the IDO pathway and lead to subsequent buildup of neurotoxic tryptophan metabolites in the developing infant. That CHEU infants display perturbed biomarkers of monocyte activation, including elevations in interferon-inducible cytokines, is supportive of this possibility[15–18]. Careful study of tryptophan replacement in the setting of chronic viral inflammation in animal models will be necessary to ensure that replacement will result in beneficial effects. If replacement alone is insufficient, then replacement with agents to modulate the kynurenine pathway, such as IDO-inhibitors, which are in development, may be necessary[31].

The primary strength of this investigation is the longitudinal profiling of almost 1600 milk samples from a deeply characterized cohort from a randomized clinical trial. The dense and consistent sampling over the first 18 months postpartum allows us to identify both temporally specific and persistent effects of maternal HIV infection. Much of the present study focuses on the first 4 months postpartum, which represents a critical window for infant development as well as the primary period of exclusive breastfeeding, and we also observed robust longitudinal perturbations in the milk metabolome. That this large study utilizes samples taken in the pre-antiretroviral therapy era is both a blessing and limitation. It is often difficult to deconvolute the profound metabolic signatures of different antiretroviral agents and HIV infection itself, which makes analysis of pure disease signatures more complex in the setting of antiretroviral therapy. However, the strength and robustness of the signatures for HIV infection observed in the present study are likely to some extent due to the severity of disease progression in the absence of effective antiretroviral therapy; maternal CD4 counts, viral loads, and tragically, mortality are all abundant evidence to this point. Growing evidence suggests that antiretroviral therapy reduces but does not fully ameliorate the chronic inflammation associated with HIV infection. Neither, unfortunately, does maternal antiretroviral therapy fully normalize the adverse health outcomes of CHEU. Consistent with these points, the metabolomic signature in the milk of WLWH is partially observed in the validation cohort of women on antiretroviral therapy with higher CD4 counts. Finally, the present study only includes samples from WLWH and WWoH with no measurements in the infants. Consequently, the effect on the infants has to be inferred as it was not directly measured. Replication of these findings in a large cohort of WLWH and their infants in contemporary settings with universal maternal antiretroviral therapy is necessary to confirm the hypothesized mechanisms on the health outcomes of CHEU.

We identified a robust and persistent metabolomic signature in the milk of WLWH comprising the kynurenine/tryptophan axis as well as dimethylarginine, cytosine, and ddhC, the free base of a recently discovered antiviral molecule, ddhCTP, that spans the first 18 months of lactation. This signature is consistent with a model in which chronic viral inflammation associated with maternal HIV infection leads to induction of the type I interferon response. Virally induced elevation of type I interferons decreases transport of tryptophan across the gastrointestinal wall, thereby leading to systemic depletion of tryptophan and subsequent reduction of tryptophan levels in the milk of WLWH. Since human milk is the only source of nutrition for exclusively breastfed infants, maternal tryptophan deficiency and chronic immune activation during gestation and lactation may drive subsequent serotonin deficiency and altered tryptophan catabolism in children of women living with HIV. If alterations in tryptophan and tryptophan catabolism are directly confirmed in

infants, this may provide opportunities for targeted interventions to ameliorate the substantial global burden of immunologic, neuro-cognitive, and growth abnormalities observed in CHEU infants.

## Methods

### Study Design

**Clinical Trial Group and parent trial study intervention.** The Zambia Exclusive Breastfeeding Study (ZEBS)[46] conducted in Lusaka, Zambia between 2001 and 2008 (clinicaltrials.gov ID NCT00310726) investigated a feeding strategy for WLWH recommended by the World Health Organization at that time. The strategy, intended to decrease infant infection with HIV during an era prior to the widespread availability of antiretroviral medications, was to rapidly wean infants from breastfeeding after 4 months of lactation. In ZEBS, WLWH were randomized to either breastfeed until usual breastfeeding cessation or to rapidly wean at 4 months. Almost all women received only a single dose of nevirapine for the prevention of vertical transmission, as was the standard of care at the time. Mother/infant pairs were followed for 24 months postpartum. From this parent trial, a sub-study of WLWH and WWoH was selected based on sample availability as described below. The clinical characteristics of the women included in this sub-study are given in Table 1.

**Sample collection.** Whole breast milk and plasma samples were stored at −80C as previously published[61].

**Sample selection and comparison groups.** Breast milk sample selection for the global metabolomics panel.

Longitudinal breast milk samples spanning the first 24 months of lactation from women living without ($N = 38$) and with HIV ($N = 288$) were selected on the basis of sample availability. Samples were selected from all 38 women living without HIV (WWoH) included in the trial, 35 of their infants survived (279 samples), and 3 infants died (9 samples). A stratified sample of WLWH was selected for inclusion by known infant outcomes for this metabolomics study. Specifically, the samples selected for study inclusion were as follows:

- All WWoH ($N = 38$; 288 samples)
- A random sample of 76 (out of 622) WLWH with infants who did not experience HIV infection and who survived ($N = 76$; 396 samples)
- All WLWH whose infants did not experience HIV infection but died in the first 24 months of life ($N = 78$; 193 samples)
- All WLWH whose infants tested positive for HIV between 1 week and 1 month of life (N = 53; 221 samples) or after 1 month of life ($N = 81$; 401 samples).

To account for the parent trial study intervention, only milk samples from women who reported they were actively still breastfeeding were included in the final analysis.

**Breast milk and plasma sample selection for the quantitative KT panel.** Breast milk (n = 101) and plasma samples ($n = 125$) were run for quantitative KT analysis. A random subset of 21 women without HIV and 80 women living with HIV, 20 each from the groups above, with paired maternal plasma and milk samples at the 4-month timepoint, were selected for the comparison. The 4-month timepoint was selected because this was the first plasma collection during lactation.

**Haiti validation cohort.** Milk samples from WWoH (n = 25) and WLWH (n = 22) from a previously described cross-sectional study were used as an independent validation cohort[48]. In contrast to the main study cohort, these women were primarily on antiretroviral therapy and had relatively minor disease progression, most with CD4 counts above 350.

**Metabolomics on the ZEBS Cohort.** Global metabolomics was performed on 1599 whole human milk samples collected longitudinally using ultra high-performance liquid chromatography/tandem mass spectrometry by Metabolon Inc. according to published methods[62–64].

**Sample Preparation.** Samples were prepared using the automated MicroLab STAR® system from Hamilton Company. Several recovery standards were added prior to the first step in the extraction process for QC purposes. To remove protein, dissociate small molecules bound to protein or trapped in the precipitated protein matrix, and to recover chemically diverse metabolites, proteins were precipitated with methanol under vigorous shaking for 2 min (Glen Mills Geno-Grinder 2000) followed by centrifugation. The resulting extract was divided into multiple fractions: two for analysis by two separate reverse phase (RP)/UPLC-MS/MS methods with positive ion mode electrospray ionization (ESI), one for analysis by RP/UPLC-MS/MS with negative ion mode ESI, one for analysis by HILIC/UPLC-MS/MS with negative ion mode ESI, while the remaining fractions were reserved for backup. Samples were placed briefly on a TurboVap® (Zymark) to remove the organic solvent. The sample extracts were stored overnight under nitrogen before preparation for analysis. *QA/QC:* Several types of controls were analyzed in concert with the experimental samples: a pooled client matrix (CMTRX) sample generated by taking a small volume of each experimental sample served as a technical replicate throughout the data set; extracted water samples served as process blanks; and a cocktail of QC standards that were carefully chosen not to interfere with the measurement of endogenous compounds were spiked into every analyzed sample, allowed instrument performance monitoring and aided chromatographic alignment. Four technical replicates ($n = 4$) of either client matrix (CMTRX) or QC samples are run for every 36 study samples in the global metabolomics panel. For the ZEBS Cohort 1599, human milk samples ($n = 1$) were analyzed with an additional 180 QC samples.

**UltraHigh Performance Liquid Chromatography-Tandem Mass Spectroscopy (UPLC-MS/MS).** All methods utilized a Waters ACQUITY ultra-performance liquid chromatography (UPLC) and a Thermo Scientific Q-Exactive high resolution/accurate mass spectrometer interfaced with a heated electrospray ionization (HESI-II) source and Orbitrap mass analyzer operated at 35,000 mass resolution[63]. The dried sample extracts were then reconstituted in solvents compatible to each of the four methods. Each reconstitution solvent contained a series of standards at fixed concentrations to ensure injection and chromatographic consistency. One aliquot was analyzed using acidic positive ion conditions, chromatographically optimized for more hydrophilic compounds (PosEarly). In this method, the extract was gradient eluted from a C18 column (Waters UPLC BEH C18-2.1 × 100 mm, 1.7 μm) using water and methanol, containing 0.05% perfluoropentanoic acid (PFPA) and 0.1% formic acid (FA). Another aliquot was also analyzed using acidic positive ion conditions; however, it was chromatographically optimized for more hydrophobic compounds (PosLate). In this method, the extract was gradient eluted from the same aforementioned C18 column using methanol, acetonitrile, water, 0.05% PFPA, and 0.01% FA, and was operated at an overall higher organic content. Another aliquot was analyzed using basic negative ion optimized conditions using a separate dedicated C18 column (Neg). The basic extracts were gradient eluted from the column using methanol and water, however, with 6.5 mM Ammonium Bicarbonate at pH 8. The fourth aliquot was analyzed via negative ionization following elution from a HILIC column (Waters UPLC BEH Amide 2.1 × 150 mm, 1.7 μm) using a gradient consisting of water and acetonitrile with 10 mM Ammonium Formate, pH 10.8 (HILIC). The MS analysis alternated between MS and data-dependent MS$^n$ scans using dynamic exclusion. The scan range varied slightly between methods but

covered 70–1000 m/z. Raw data were uploaded to a publicly available database; see data availability below. *Data Analysis:* Compounds were identified by comparison to library entries of purified standards or recurrent unknown entities based on authenticated standards that contain the retention time/index (RI), mass to charge ratio (*m/z*), and fragmentation data. Biochemical identifications are based on three criteria: retention index within a narrow RI window of the proposed identification, accurate mass match to the library +/− 10 ppm, and the MS/MS forward and reverse scores between the experimental data and authentic standards. *Metabolite Quantification and Data Normalization:* Peaks were quantified using area-under-the-curve. A data normalization step was performed to correct variation resulting from instrument inter-day tuning differences. Essentially, each compound was corrected in run-day blocks by registering the medians to equal one (1.00) and normalizing each data point proportionately.

**Metabolomics on the Haiti Cohort.** Frozen human milk samples from the Haiti Cohort were processed by Metabolon, Inc. using the CMTRX made from the ZEBS Cohort in order for the data to be merged. Instrument variability was determined by calculating the median relative standard deviation (RSD) for the internal standards that were added to each sample prior to injection into the mass spectrometers. Overall process variability was determined by calculating the median RSD for all endogenous metabolites present in 100% of the Client Matrix samples, which are technical replicates of pooled client samples. For the Haiti Cohort 47 human milk samples ($n = 1$) were analyzed with an additional 8 QC samples.

**Identification of ddhC.** The independent identification of ddhC was conducted at the Northwest Metabolomics Research Center, University of Washington. Breast milk samples were processed using a protein precipitation method. Briefly, 250 μL methanol was added to 50 μL breast milk, vortexed, stored at −20 °C for 20 min, centrifuged at 18,000 x $g$ for 15 min at 4 °C, and 150 μL of the supernatant was collected. Samples were dried in a Vacufuge at 30 °C and reconstituted in 500 μL of LC-matching solvent before LC-MS analysis.

MS analysis was performed using an Agilent 6546 LC/Q-TOF MS using HILIC chromatography. Accurate mass analysis identified the molecular formula of X-12127 as $C_9H_{11}N_3O_4$, based on the [M + H]+ ion. Tandem MS/MS with a collision energy ramp of 5–40 V revealed its fragmentation patterns. While no database matches were found, the presence of a cytosine fragment suggested the metabolite might be ddhC. This identification was confirmed by matching the experimental MS/MS spectrum and retention time of [M + H]+ with a ddhC standard, and the identification was finalized using a parallel analysis of a ddhC reference standard with the study samples.

**Quantitative KT Panel.** Plasma and whole breast milk samples were analyzed for Tryptophan and Kynurenine by LC-MS/MS (Metabolon Method TAM220: "LC-MS/MS Method for The Determination of 15 Metabolites in Human Plasma (GFR-Panel)". The method was adapted to analyze whole breast milk samples. Briefly, the plasma and whole milk samples were extracted with an organic solvent containing isotopically labeled internal standards (Tryptophan-$d_5$, CIL Catalog# DLM-1092-0 and Kynurenine-$d_6$, CIL Catalog# DLM-7842-0), injected on the Agilent 1290/SCIEX QTrap 5500 LC MS/MS system with a BEH Amide UHPLC column operating in positive mode using electrospray ionization, and the peak area of each analyte product ions measured against the peak area of the product ions of the corresponding internal standards. Quantitation was performed using a weighted linear least squares regression analysis generated from fortified calibration standards with ranges of 0.0250-10.0ug/mL for L-Kynurenine and 0.200-80.0ug/mL for L-Tryptophan that were generated immediately prior to the run (Tryptophan, Sigma Catalog#

**Table 2 | R packages used in the analyses**

| Package | Version | Use |
|---|---|---|
| vegan | 2.6-2 | PERMANOVA |
| lmerTest | 3.1-3 | Linear mixed effects models |
| lme4 | 1.1-30 | Linear mixed effects models |
| emmeans | 1.8.1-1 | Estimated marginal means of linear mixed effects models |
| ranger | 0.13.1 | Random forests analyses |

93659 and Kynurenine, Sigma Catalog# 61250). For the quantitative/targeted KT panels, 6 QC samples are run per batch. These are spiked samples with varying concentrations of analyte stocks for Tryptophan and Kynurenine. Two are low concentration, two are medium concentration, and two are high concentration. There were 101 milk samples and 125 plasma samples analyzed, with an additional 24 QC samples for this study.

**Analysis.** Principal coordinates analysis and permutational multivariate analysis of variance (PERMANOVA) were performed using Euclidean distances calculated on normalized metabolite abundances. Linear mixed effects models of the form metabolite ~ HIVExposure*Visit + (1 | PID), where HIVExposure is an indicator variable for WLWH versus WWoH, Visit is a categorical variable for the study visit, and PID is the unique participant identifier, were used to identify differentially abundant metabolites as well as differences in KT log-ratio. Estimated marginal means were used to conduct the post hoc comparisons, either averaged or stratified by study visit. Results are reported as t-ratios, which are the estimates of the regression coefficients divided by their standard error. For the quantitative KT panel data, comparisons were performed using a Kruskal-Wallis (for more than two categories) or Welch's t (for two categories) test. Spearman's rank correlation was used to compare global and quantitative KT data and to compare metabolite abundances with maternal CD4 counts and viral loads.

Random forest classification models were constructed separately for each study visit with HIV exposure as the outcome. A two-step approach was utilized to select both the optimal number of features as well as the specific features used for each model. In the first step, one hundred forests each comprising 10,000 trees were built to obtain feature importance values calculated as mean decrease in accuracy. In the second step, tenfold cross-validation with a sequentially reduced number of features was then used to identify the optimal number of features to be used. Finally, a sparse model was constructed for each study visit containing the optimal number of metabolites calculated in step 2 (up to a maximum of twenty features to aid interpretability), with features selected by the highest importance as calculated in step 1. Model accuracy, sensitivity, specificity, and Matthew's correlation coefficient were calculated from the out-of-bag error estimate for the final sparse model.

Analysis of the Haiti validation cohort was performed using linear regression models of the form metabolite ~ HIVExposure + InfantAge, as there were no repeated measures. Infant age in days was included as a covariate in this regression model, given the range of ages at which milk samples were collected. Regression estimates from this validation cohort were compared to those from the main cohort at the 1-month and 4-month timepoints, as this best matched the distribution of infant ages between the two cohorts.

All statistical analyses were conducted in the R statistical environment (version 4.1.3). A full list of all R packages used in the analyses is given in Table 2. All p-values were adjusted for multiple comparisons using the Benjamini-Hochberg false discovery rate method, and a cutoff of $p < 0.05$ was accepted as significant.

**Ethics Approval and consent to participate**

All women provided written informed consent. The original protocol was approved by institutional review boards at Boston University, Children's Hospital of Los Angeles, Columbia University, and the University of Zambia Research Ethics Committee. The current study was approved by Children's Hospital of Los Angeles (IRB# CCI-03-110-CR002), University of California, Los Angeles (IRB# 16-000798), and Columbia University (AA6449 and AJ8851).

**Reporting summary**

Further information on research design is available in the Nature Portfolio Reporting Summary linked to this article.

## Data availability

The MS data generated in this study have been deposited in the MetaboLights database under accession codes MTBLS2307 and MTBLS2573.

## Code availability

The analysis code is available at GitHub[65] [https://github.com/fanli-gcb/BMM] and Zenodo [https://doi.org/10.5281/zenodo.16658313].

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

## Acknowledgements

We would like to acknowledge and thank the study participants and the ZEBS and Haiti Study Teams. This work received funding from the National Institutes of Health, National Institute of Child Health and Human Development NICHD R01 HD 39611 (LK, DT), NICHD R01 HD 40777 (GMA), NICHD 1-R01 HD 057617 (LK, GMA), NIH R21 HD 49287 (JS), Doris Duke Charitable Foundation CU52209601 (LK), PEPFAR GHS-A-00-00020-00 (LK) and NIH P30DK035816 (WZ, DR). The sponsors had no role in the study design. Overall support for the International Maternal Pediatric Adolescent AIDS Clinical Trials Network (IMPAACT) was provided by the National Institute of Allergy and Infectious Diseases (NIAID) with co-funding from the Eunice Kennedy Shriver National Institute of Child Health and Human Development (NICHD) and the National Institute of Mental Health (NIMH), all components of the National Institutes of Health (NIH). under Award Numbers UM1AI068632 (IMPAACT LOC), UM1AI068616 (IMPAACT SDMC) and UM1AI106716 (IMPAACT LC), and by NICHD contract number HHSN275201800001I. The content is solely the responsibility of the authors and does not necessarily represent the official views of the NIH.

## Author contributions

N.H.T., G.M.A., L.K. and F.L. conceived the study, designed the research, and coordinated the project. N.H.T., F.L., K.G.F., L.K., J.S. and G.M.A. interpreted the data, and N.H.T. and F.L. drafted the primary draft of the manuscript. W.Z. and D.R. performed the experiments to identify X-12127, helped interpret the data, and drafted the relevant sections of the manuscript. FL performed the statistical analyses. All authors reviewed and approved the final manuscript.

## Competing interests

The authors declare no competing interests.

### Consent for publication

All authors approved this manuscript for publication.
