## [Peer Review file · Nature Communications]

Altered milk tryptophan and tryptophan metabolites in women with HIV

Corresponding Author: Grace Aldrovandi

Version 0:

Reviewer comments:

Reviewer #1

(Remarks to the Author)

Please see the attached comments.

(Remarks on code availability)

Reviewer #2

(Remarks to the Author)

This is an interesting study which profiled the milk metabolome over 18 months postpartum in a group of women with and without HIV who were enrolled to a Zambian randomized breastfeeding intervention of early weaning versus usual duration, in the pre-ART era. The authors report global metabolomic differences between groups and focus in on specific reductions in tryptophan and altered tryptophan catabolism, plus elevation of a novel compound (ddhC). There is a clear need to explore the metabolomics of milk in women with HIV to evaluate its contribution to poor health outcomes of infants who are HIV-exposed. Overall, this is therefore a valuable contribution to the field, but there are several major limitations in the cohorts leveraged, and a general over-interpretation of the findings.

Major comments

1. The ZEBS cohort included a randomized intervention that has not been accounted for in the analysis (beyond excluding samples that were collected post-weaning) and had a mortality of over 50% in the infants of women with HIV. Together, these factors lead to a major decline in sample numbers over time – presumably because infants die (so their mothers are not breastfeeding) or because those randomized to early weaning have stopped breastfeeding. Both these factors need to be accounted for in the analysis. The high mortality in particular is not mentioned in the results section, beyond a comparison in the Table which has an asterix (not clear why?) but no p value comparison. This feels like a key oversight because these findings may simply reflect very sick women whose infants are also sick and experience high mortality due to untreated HIV.
2. The selection of samples was difficult to ascertain from the methods section – in particular, whether both trial arms were sampled equally and how they accounted for mortality among infants with HIV, in selecting mother-infant pairs. How many time-points did each mother contribute? The reader needs to appreciate who is in the cohort over time, and how many repeated measures were included. How different were the women in both groups who provided later samples, compared to those who provided only early samples (either because they subsequently weaned or because their infant died)? Overall, how have the authors accounted for this inevitable missing data over time in the analysis?
3. Because of the challenges for the reader in following the sampling strategy, a flow chart would be very helpful, to show the original ZEBS cohort (which presumably included women without HIV who were randomized also?), split by randomized arm, and sample collection (during the trial) and availability (following storage), and how different groups were then selected for this study. Similarly, figure 1 would benefit from some more information to show excluded samples (ie. how the 1599

samples declined to 1426).

4. The authors state that the targeted KT analyses were done at 4 months because almost all women had available breast milk samples, and there were paired plasma samples available. But only 21/32 women without HIV (66%) and 77/189 (40%) of women with HIV who had untargeted metabolomics done, also had targeted KT analysis. Why was this not done in all women, and can the authors show a supplementary table to show how similar or different the women were who had versus did not have the targeted analysis done, so readers can appreciate whether this subgroup of women were similar or different to the whole cohort.

5. The linear models do not include any covariates that might be important, such as maternal age, nutritional status (eg MUAC) and infant sex. These are presumably all factors that might contribute to different metabolic profiles of milk?

6. The authors state that the Haiti cohort validates their findings in a less sick cohort. It is definitely a strength that this study had access to another cohort, albeit small, but I do not see evidence that the findings were replicated. Why is the volcano plot for the Haiti cohort not shown? Did the authors adjust for the difference in infant age between women with and without HIV, given that this was an important determinant of metabolic profiles in the larger cohort from Zambia? The only finding that is replicated with a significant adjusted p value is the elevation in cytosine. Given that this is actually the key finding across cohorts, it is not clear why the paper focuses more on KT ratios than on cytosine, and why this key (replicated) difference was not taken forwards more for further exploration? None of the adjusted P values for tryptophan or KT ratio were significant in the Haiti cohort. It is confusing in the results section to switch between adjusted and unadjusted P value results to explain the findings of the cohort.

7. Were plasma samples available from infants? It would be very intriguing to see if lower tryptophan levels in breast milk led to lower circulating plasma tryptophan and altered metabolic profiles in the infants.

8. Overall, the discussion section makes quite definitive claims. The results are novel, and intriguing, but the authors need to be clear that they were not replicated in a less sick cohort, and these findings may therefore simply reflect a very sick cohort of patients with advanced HIV disease, no access to ART, and 50% child mortality. A large, sub-Saharan African cohort, who are stable on ART, is really needed now to replicate these initial findings. The mechanistic explanation for the findings presented in Figure 5 feels like quite a leap, because it combines cytokine alterations (not explored in this study), with other metabolites not examined (I could not see evidence of serotonin or quinolinic acid measurements in this cohort, for example), to pull together a mechanistic explanation. The authors need to acknowledge that this study was limited in its ability to explore mechanism and Figure 5 therefore feels extremely speculative.

9. Overall, the discussion section needs to better acknowledge the limitations of the analysis (e.g. missing data, multiple subgroups, non-replicated findings in the smaller Haiti cohort) and make somewhat less definitive claims (eg the conclusion speculates that infants may have serotonin deficiency and altered tryptophan catabolism, but these were not even explored).

Minor comments

1. Please add some sociodemographic data to Table 1 to contextualise the cohorts (eg maternal education, some marker of SES, rural versus urban) as well as nutritional status and HIV disease stage.

2. Did women with recent infections or mastitis get excluded from analysis, as these may have altered breast milk metabolites.

3. I am surprised to see no authors from the countries where the cohorts were recruited.

(Remarks on code availability)

Reviewer #3

(Remarks to the Author)

I would like to thank the authors for submitting their manuscript and for addressing an important topic. The manuscript investigates a critical knowledge gap, in understanding the human milk metabolome of women with HIV (WWH) compared to women without HIV (WWoH). The study profiled the milk metabolome of WWH and WWoH and identified decreased and increased levels of tryptophan and kynurenine in WWH compared to WWoH as well as a high tryptophan: kynurenine ratio in WWH.

However, after careful consideration, I have several major concerns that, in my view, significantly limit the suitability of the manuscript for publication in its current form.

1. Title: The study primarily focused on the milk metabolome. There is no analysis or results to show how the findings relate to the health of children born to women with HIV.

2. Abstract: "Our findings provide intriguing evidence that decreases in tryptophan availability and perturbations in tryptophan catabolism in children born to WWH may contribute to their increased morbidity and mortality". While this is a promising hypothesis, the study did not test this and giving this statement feels a bit of an overreach.

3. Introduction:

- a. The authors indicated that they used a healthier cohort of WWH from Haiti for validation. “Lastly, we confirm this milk metabolomic signature of HIV in a second, healthier cohort of WWH on antiretroviral therapy”. How this healthy cohort was defined is not clear.
- b. The authors have not indicated in the introduction whether the tryptophan level in milk changes over time during the lactation period. This would enrich the readers to understand tryptophan fluctuations during infancy given they conducted a longitudinal study through 18 months of infancy.

4. Methods:

- a. The current study analysed 228 WWH and 36 WWoH selected from a clinical trial. The authors did not provide the number of participants enrolled in the original study. It's not known what fraction of the participants from the clinical trial were analysed in the current study. Is the proportion of participants analysed in this study reflective of the original trial design?
- b. The only selection criterion indicated was that the WWH had not ceased breastfeeding at 4.5 months, however, it is unclear whether these mothers continued to breastfeed until the end of the observation period which is 18 months. This is critical as breastfeeding itself changes milk output and its composition.
- c. The data was collected in the pre-antiretroviral (ART) era where ART coverage had not widely spread, however, the authors indicated that most of the women in the current analysis received a single dose of nevirapine for prevention of vertical transmission per standard of care. Authors could provide the percentages of WWH who received nevirapine because nevirapine usage alone can alter the milk metabolome.
- d. Among the criteria for exclusion and inclusion of samples was the weaning period i.e., samples collected post-weaning period were excluded (n = 99). However, the authors did not define the weaning period in this population. Additionally, the readers would benefit if the authors could provide a concise consort on the exclusion and inclusion of participants for different analyses done in the current study.

5. Results:

- a. Table 1 presents the baseline characteristics of the study participants including continuous variables such as maternal age, CD4 and CD8 counts, and infant birth weight. It is confusing whether these continuous variables were summarised as median (interquartile ranges [IQR]) or mean (standard deviations [SD]). Currently, the authors presented them as median (SD) which is confusing. This is also seen in Extended Data Tables and Methods. Please clarify this. Median is a non-parametric measure, whereas SD is a parametric measure. They do not naturally combine.
- b. Table 1: what is the unity of measure for parity?
- c. “WWH had lower CD4 T cell counts (315 vs. 840 cells/mm³) and higher CD8 T cell counts (759 vs 563 cells/mm³) than WWoH (p<0.001 and p=0.001, respectively)”. The these values in the text don't match the values indicated in Table 1 under maternal CD4 and CD8 count.
- d. Table 1: for consistency, the authors could indicate the p-value for the difference in gestational age between WWoH and WWH. For the rest of the variables in the table, the authors computed their p-values
- e. Table 1: See the below excerpt of Table 1. What does asterisk (*) mean? You have indicated that the p-value is NA. Should this be the case?

Value WWoH WWH p-value

Child death (%) 3 (7.9) 145 0.3)* NA

- f. The authors indicated that broadly the milk metabolic profiles were strongly influenced by study visits, HIV infection, maternal CD4 count, and infant sex. “Overall milk metabolomic profiles were strongly influenced by study visit (4.0% variance, p<0.001), and by HIV infection (0.2% variance, p<0.001), maternal CD4 count (0.15% variance, p<0.001), and infant sex (0.07% variance, p=0.009) (Extended Data Fig. 1)”. The cited Extended Data Fig. 1 by the authors shows PC1 and PC2 of metabolite clustering coloured by maternal HIV status (WWH and WWoH). From this figure, there is no clear separation of metabolites by maternal HIV status as the ellipses showing 95% confidence areas of WWH and WWoH greatly overlap. Further, there is no information about study visits, maternal CD4 count, and infant sex on metabolomic variance in this Figure as cited by the authors. The reported variances and their corresponding significance levels are not provided by the authors in Extended Data Fig. 1

- g. Authors reported that linear mixed effect regression identified 173 metabolites significantly altered in WWH compared to WWoH across all the study visits and cited Extended Data Table 1. However, from Extended Data Table 1 at an adjusted p-value of <0.05 as indicated in the methods, 104 metabolites and not 173 metabolites were significant. Could the authors clarify this discrepancy?

- h. “Milk levels of tryptophan were 6.8% and 8.3% of the plasma levels for WWoH and WWH, respectively; this proportion did not differ significantly by maternal HIV infection (p=0.14, Extended Data Fig. 8A and Extended Data Table 8). A similar trend was observed for kynurenine with breast milk levels at 32.4% and 36.6% of the corresponding plasma levels for WWoH and WWH, respectively (p=0.42, Extended Data Fig. 8B and Extended Data Table 8)”. Here, the authors are reporting results of the Milk/Plasma ratio for tryptophan or kynurenine, which according to the authors' definition investigated the transfer of tryptophan or kynurenine from the bloodstream to milk. This computation gives a “ratio” which may not directly translate to proportions. The authors reported proportions while they cited Extended Data Figure 8A which indicates the boxplots of concentration of tryptophan and kynurenine in plasma and milk samples stratified by maternal HIV status. These are median values and not proportions. In Extended Data Table 8 the author reported tryptophan or kynurenine availability computed by Milk / Plasma Ratio as shown below. Could the authors explain whether this ratio directly translates to proportion and its interpretation?

Table 8. Quant KT

WWH WWoH p-value

Tryptophan, median (sd)

Milk 0.318 (0.177) 0.620 (0.222) 0.00082

Plasma 5.110 (2.357) 8.380 (2.298) 1.29E-07

Milk / Plasma Ratio 0.083 (0.038) 0.068 (0.025) 0.14

Kynurenine, median (sd)

Milk 0.147 (0.117) 0.109 (0.084) 0.028
Plasma 0.453 (0.158) 0.362 (0.115) 0.00065
Milk / Plasma Ratio 0.366 (0.220) 0.324 (0.199) 0.42

i. The random forest results do not contribute anything to the paper.

j. The study is limited to the investigation of free tryptophan. However, essential amino acids could also be incorporated into albumin and other proteins in the breastmilk, which will be digested by the infant. Arguing that the lower free tryptophan levels in this study could equate to having poor health outcomes to the child is not supported by data.

(Remarks on code availability)

Reviewer #4

(Remarks to the Author)

(Remarks on code availability)

Version 1:

Reviewer comments:

Reviewer #1

(Remarks to the Author)

In this revised manuscript, Tobin et al. have fully addressed all previously suggested changes, substantially improving the clarity and emphasis on the study's longitudinal design and validation approach. The authors now more effectively highlight the depth and nuance of their rich dataset, particularly in relation to (a) the longitudinal validation of tryptophan dysregulation in women with HIV, (b) the annotation of 3'-deoxy-3'-4'-dihydro-cytidine (ddhC), and (c) the cross-cohort biomarker validation using samples from Haiti. These refinements further underscore the significance of their findings and the strength of the study's design. The manuscript now presents a clear, compelling, and rigorously supported narrative on the impact of HIV on maternal and infant health via breast milk. I recommend acceptance of this manuscript for publication in Nature Communications.

(Remarks on code availability)

Reviewer #2

(Remarks to the Author)

The authors have responded to my queries adequately

(Remarks on code availability)

Reviewer #3

(Remarks to the Author)

I appreciate the efforts made by the authors in revising the manuscript based on the reviewers' suggestions.

However, I still believe that the abstract (albeit unintentionally) overrepresents the results. The statement "Our findings suggest that decreases in milk 59 tryptophan availability and perturbations in milk tryptophan catabolism in WWH may contribute to the increased morbidity and mortality of children born to WWH." appears to have been revised, but actually still means the same. The study does not suggest anything about the morbidity or mortality of children, neither was it measured directly or indirectly. I suggest the following text: "Our findings demonstrate that milk tryptophan content and availability decreases among WWH, which may indicate perturbations in milk tryptophan catabolism. The link between this perturbation and the increased morbidity and mortality of children born to WWH merits further investigation."

My opinion about the random forest results still stand. I do not believe that it adds anything to the manuscript - the results are just as strong without it. In the answer of the authors to the comment, they said: "We respectfully disagree with the Reviewer on this point. The random forests results provide confirmatory evidence for the linear regression analyses identifying key metabolites that differentiate WWH and WWoH. Furthermore, as random forests is a nonlinear approach, it adds value in showing that both linear and nonlinear methods identify the same set of key metabolites that are perturbed in the milk of WWH".

Random forest does not provide the best confirmatory evidence for the association between tryptophan and WWH. The

confirmatory evidence you need is to see the same association in a different cohort, which you have. Furthermore, the authors argue that RF is a non-linear approach, showing both linear and non-linear methods agree with the results. This is not entirely correct. Random forest is ABLE TO hand/uncover non-linear associations, but the actual model it fits could also be linear. There is no telling whether the model fit a linear or non-linear association in your data.

The results are just as good without the random forest model. I know it is fancy, but not needed. I refer to the decision of the editor about this issue.

On my remark about having only free tryptophan analysed in the study, the authors responded: "it is only free tryptophan that is the form that crosses the blood-brain barrier for the production of brain serotonin". I agree. BUT, free tryptophan could also be formed from the bound tryptophan during gastrointestinal digestion. This should be considered as well in the manuscript.

The authors further answered: "It is not known if the low levels of tryptophan in the milk are still sufficient for the infants, how well the tryptophan is absorbed by the infant in the setting of exposure to chronic viral inflammation, or if the systemic inflammation which has been characterized in CHEU infants leads to alterations in tryptophan catabolism with an increase in neurotoxic metabolites. All of these possibilities deserve careful study." I propose to also include this sentence in the manuscript.

(Remarks on code availability)

Reviewer #4

(Remarks to the Author)

(Remarks on code availability)

The codes are available, but access to the data requires a private link which I do not have. Therefore, I have not ascertained if the codes and results are reproducible.

Reviewer #1 (Remarks to the Author):

In this manuscript, Tobin et al. present an exceptional longitudinal study of breast milk samples for women with HIV (WWH) and women without HIV (WWoH) through an initial investigation of a cohort in Zambia that was validated by samples collected from a cohort in Haiti. Through this work, the authors were able to make several significant strides in elucidating the effects of HIV on infants through breast milk. These include the (a) longitudinal validation of tryptophan dysregulation in WWH, (b) the annotation of 3'-deoxy-3'4'-didehydro-cytidine (ddhC), and (c) the validation of biomarkers in a separate Haitian cohort. Collectively, the text and figures together tell a clear and compelling story of HIV's effects on maternal and infant health. However, the unique insights from the longitudinal sampling and validation study currently feel underemphasized. This reviewer recommends major revisions to better highlight the depth and nuance of this rich dataset. With these revisions, the manuscript would be a strong contribution to Nature Communications and its broad readership.

Revisions and Comments:

1. *Figure images appear pixelated in the rendered pdf and should be reviewed before publication.*

We thank the Reviewer for noting the pixelated figures in the PDF. We have fixed this in the revised manuscript. We learned that the default option in Microsoft Word is to compress images in saved files, which leads to pixelation when converted to PDF and have now corrected this.

2. *Can the authors clarify why the 1- and 4-month time points were highlighted in the Zambia cohort? Are these associated with specific biological or physiological events? Other time points appear to show more pronounced molecular changes, so the focus on these two should be justified in the text.*

We focused on the 1- and 4-month time points in the ZEBS cohort because these occur at a point when breastfeeding is established as the sole source of nutrition for the infant and also reflect a critical developmental window for the infant. However, the Reviewer has a good point in that other metabolites that we did not focus on will likely be of interest to many readers. We added additional supplementary longitudinal data in the results section: "Other features, including many lipid and amino acid metabolites, were found to be altered later in lactation (**Supplementary Fig. 3** and **Supplementary Data 1**) and may reflect the far-reaching impact of HIV infection on milk composition."

Additionally, we have added the following text to the Discussion:

"Much of the present study focuses on the first 4 months postpartum, which represents a critical window for infant development as well as the primary period of exclusive breastfeeding, and we also observed robust longitudinal perturbations in the milk metabolome."

3. *In Figure 2, 3 and several supplemental figures, the authors show normalized abundances of several molecules across time points with black and red stars on each plot. The meaning of these stars is unclear and should be specified in the text.*

The meaning of the red and black stars in these figures is described in the legends:
For **Figure 2**:

"Red asterisks along the top denote study visits at which the selected compound was

differentially abundant in WWH versus WWoH. Black asterisks along the bottom denote study visits at which the compound was selected as a predictive feature in the random forests modeling.”

For **Figure 3**:

“* $p < 0.05$, ** $p < 0.01$, *** $p < 0.001$.”

For **Supplementary Figure 6** (updated numbering is now Supplementary Figure 7):

“Red asterisks along the top denote study visits at which the selected compound was differentially abundant in WWH versus WWoH. Black asterisks along the bottom denote study visits at which the compound was selected as a predictive feature in the random forests modeling.”

For **Supplementary Figure 7** (updated numbering is now Supplementary Figure 9):

“* $p < 0.05$, ** $p < 0.01$, *** $p < 0.001$.”

4. *It is recommended that Figure 1E be amended to include the timeline of sample collection for the Haiti validation cohort. Additionally, a justification for comparison to the 4-month timepoint from the Zambia cohort could be provided within the methods text.*

We thank the Reviewer for this helpful suggestion. We have added a diagram to **Figure 1E** to show the timing of sample collections for the Haiti validation cohort.

We have also expanded the comparison of the Haiti regression estimates to both the 1-month and 4-month timepoints from ZEBS, as these best match the overall distribution of ages from the Haiti cohort. The text now reads:

“Analysis of the Haiti validation cohort was performed using linear regression models of the form *metabolite* ~ *HIVExposure* + *InfantAge* as there were no repeated measures. Infant age in days was included as a covariate in this regression model given the range of ages at which milk samples were collected. Regression estimates from this validation cohort were compared to those from the main cohort at the 1 month and 4 month timepoints as this best matched the distribution of infant ages between the two cohorts.”

We have also updated the comparisons of the ZEBS and Haiti cohorts as shown in

Supplementary Figure 11A. The relevant part of the Results section now reads:

“Regression estimates of the associations between HIV status and metabolites were obtained while including infant age in days as a covariate given the non-uniformity of sampling. Nevertheless, estimates from the Haiti cohort were strongly correlated to those from the main cohort at the 1 month and 4 month visits ($r=0.126$ and 0.386 , $p<0.001$ for all metabolites, respectively; $r=0.812$ and 0.789 , $p<0.001$ when only the metabolites observed to be significantly associated with HIV status in the ZEBS cohort were included; Supplementary Fig. 11A).”

5. *The authors mention 9,10-DiHOME and 12,13-DiHOME metabolites were significantly elevated in WWH at 1 month. These molecules are, however, isomers and indistinguishable with MS. It should be clarified in text how these molecules are being annotated as the respective isomers observed?*

We appreciate the Reviewer’s diligence in assessing the metabolomics data. To clarify, 9,10-DiHOME and 12,13-DiHOME share the same mass, but the two compounds were chromatographically resolved by their different retention times and fragmentation patterns. Furthermore, both molecules were validated against authenticated standards by Metabolon.

We note that this is described in the Methods as follows:

“Compounds were identified by comparison to library entries of purified standards or recurrent unknown entities based on authenticated standards that contains the retention time/index (RI), mass to charge ratio (m/z), and fragmentation data. Biochemical identifications are based on three criteria: retention index within a narrow RI window of the proposed identification, accurate mass match to the library +/- 10 ppm, and the MS/MS forward and reverse scores between the experimental data and authentic standards.”

There is also a detailed description of each of the 4 UPLC-MS/MS methods (PosEarly, PosLate, Neg, Polar/HILIC).

6. *Supplemental Figures 6 and 7, which appear to go alongside the KT ratio results (Figure 3) of the main text show continued significance of tryptophan (assuming stars indicate significance). Alternatively, kynurenine does not appear significant at any time point. The authors should justify in text the continued significance of the KT ratio if kynurenine is not significant*

We chose to utilize the KT ratio as it has been previously shown to be associated with IDO activity and disease progression in people with HIV (Huengsborg M et al. Clin Chem 1998)¹(Bipath P. BMC ID 2015)² (Favre D. STM 2010)³ and other populations (Mehraj, V. & Routy, J.-P.. Int. J. Tryptophan Res. 2015)⁴ (Silvano, A. et al. J. Reprod. Immunol. 2023)⁵(Collins, J. M. et al. JCI Insight, 2020)⁶(Gazi MA. Am J Trop Med Hyg 2020)⁷ (Johnson KE et al. Nat Comm. 2024)⁸. We would also like to note that kynurenine is significantly elevated in the mean effects model (**Supplementary Table 2**).

7. *The authors note that this study uniquely provides longitudinal insights into metabolic changes in breast milk from WWH and WWoH, which is certainly a strength. However, the longitudinal aspects feel underemphasized in the main text. In contrast, the supplemental figures reveal a richer temporal dimension, with several compound groups clearly identified at specific time points. These observations could be further emphasized by the authors to highlight the longitudinal insights from this work.*

We focus on the developmentally critical time window between 1 and 4 months of age as discussed in comment #2. However, we agree with the Reviewer that there are also very interesting longer term temporal components to the differences identified between WWH and WWoH. Therefore, we have added the following text to the Results along with the referenced Supplementary Data 1:

“Other features, including many lipid and amino acid metabolites, were found to be altered later in lactation (**Supplementary Fig. 3** and **Supplementary Data 1**) and may reflect the far-reaching impact of HIV infection on milk composition.”

The newly added **Supplementary Data 1** is an additional PDF file that shows line plots of metabolite abundances over the study course for all metabolites that were identified by either linear regression or random forests analyses.

This is an interesting study which profiled the milk metabolome over 18 months postpartum in a group of women with and without HIV who were enrolled to a Zambian randomized breastfeeding intervention of early weaning versus usual duration, in the pre-ART era. The authors report global metabolomic differences between groups and focus in on specific reductions in tryptophan and altered tryptophan catabolism, plus elevation of a novel compound (ddhC). There is a clear need to explore the metabolomics of milk in women with HIV to evaluate its contribution to poor health outcomes of infants who are HIV-exposed. Overall, this is therefore a valuable contribution to the field, but there are several major limitations in the cohorts leveraged, and a general over-interpretation of the findings.

Major comments

1. *The ZEBS cohort included a randomized intervention that has not been accounted for in the analysis (beyond excluding samples that were collected post-weaning) and had a mortality of over 50% in the infants of women with HIV. Together, these factors lead to a major decline in sample numbers over time – presumably because infants die (so their mothers are not breastfeeding) or because those randomized to early weaning have stopped breastfeeding. Both these factors need to be accounted for in the analysis. The high mortality in particular is not mentioned in the results section, beyond a comparison in the Table which has an asterisk (not clear why?) but no p value comparison. This feels like a key oversight because these findings may simply reflect very sick women whose infants are also sick and experience high mortality due to untreated HIV.*

We appreciate that the description of the study design and sample selection was not very clear, which is why both remarks #1 and #2 are very helpful.

Specifically on the topic of the ZEBS parent trial, the randomized intervention was for women living with HIV to be randomized to either breastfeeding for their preferred duration or to abrupt cessation of breastfeeding at 4 months. Women without HIV were not randomized and breastfed for their preferred duration. Thus, over the first 4 months, the randomization has no relevance as practices in the two randomized groups were identical. After 4 months, although samples were collected at 4.5 months in both groups to ascertain whether breastfeeding had ended (Thea DM et al. AIDS 2006)⁹, samples from those known not to still be breastfeeding at 4.5 months were excluded from all analyses. Our approach to the analysis is the methodologically sound way to analyze these data to address the study questions.

On the issue of high child mortality, we again apologize for the lack of clarity in the Methods. The apparent high child mortality reported in Table 1 is the combined mortality of the cohorts of infants selected for this analysis and is largely a reflection of the sample selection. Briefly, we selected all available WWoH (38), all available WWH whose infants died while remaining uninfected (78), and all available WWH whose children acquired HIV via intrapartum (53) or postnatal (81) infection. We also included a random sample of WWH whose children survived the study course while remaining uninfected (76 selected out of 622 total). This was described in the Methods under the “Breast milk sample selection for the global metabolomics panel:” sub-section.

However, we agree with the Reviewer that the existing description was unclear and made it difficult to ascertain the actual selection of samples. Therefore, we have amended this section to use a list format of the exact groups selected for this study. The relevant section now reads:

“Longitudinal breast milk samples spanning the first 24 months of lactation from women living without (N=38) and with HIV (N=288) were selected on the basis of sample availability. Samples were selected from all 38 women living without HIV (WWoH) included in the trial, 35 of their infants survived (279 samples) and 3 infants died (9 samples). A stratified sample of WWH were

selected for inclusion by known infant outcomes for this metabolomics study. Specifically, the samples selected for study inclusion were as follows:

- All WWoH (N=38; 288 samples)
- A random sample of 76 (out of 622) WWH with infants who did not experience HIV infection and who survived (N=76; 396 samples)
- All WWH whose infants did not experience HIV infection but died in the first 24 months of life (N=78; 193 samples)
- All WWH whose infants tested positive for HIV between 1 week and 1 month of life (N=53; 221 samples) or after 1 month of life (N=81; 401 samples).

To account for the parent trial study intervention, only milk samples from women who reported they were actively still breastfeeding were included in the final analysis.”

Regarding the description of child mortality in Table 1, we apologize for the omission. The description of the asterisk was mistakenly dropped in revisions. The asterisk and lack of p-value comparison is meant to indicate to the reader that no comparison was made because the balanced child mortality was by sampling design, not due to the randomized intervention in the parent ZEBS trial. We have added this back as a footnote in Table 1:

“* Based on sampling strategy”

We agree with the Reviewer that it is important to verify that our findings are not simply a reflection of very sick women with advanced disease whose infants are also very sick (either in terms of acquiring HIV or mortality), particularly given the sampling strategy that was used for this study. We addressed this in the analysis by performing a subgroup analysis of WWH whose children remained uninfected as noted in the Results:

“Finally, a sub-analysis of WWH whose children remained uninfected during the course of the study (38 WWoH versus 154 WWH with CHEU) confirmed the metabolic signature of maternal HIV infection to be independent of child outcome (**Supplementary Fig. 5 and Supplementary Tables 6 and 7**).”

We have now further extended this idea with a new analysis.

We repeated the comparison of WWH versus WWoH but limited the analysis to those whose children survived through the study course (in other words, excluding all women with a child mortality event). This has been added as **Supplementary Fig. 6B**, and the relevant portion of the Results now reads:

“Finally, sub-analyses of WWH whose children remained uninfected during the course of the study (38 WWoH versus 154 WWH with CHEU) and those whose children survived through the entire study course (35 WWoH versus 143 WWH) confirmed the metabolic signature of maternal HIV infection to be independent of child outcome (**Supplementary Fig. 6 and Supplementary Tables 6-9**).”

The legend for **Supplementary Figure 6** now reads:

“Sub-analysis of WWH versus WWoH among those whose children remained uninfected (a) or whose children survived the entire study course (b). Heatmap of mean importance values from random forests models of WWH versus WWoH. Only features selected by RF models at more than one timepoint are shown. Coloring indicates the mixed model regression coefficient for each timepoint-feature combination.”

Together, these analyses suggest that the drastic alterations observed in WWH are in fact related to HIV infection and are not driven by very sick women whose infants experience high mortality.

2. The selection of samples was difficult to ascertain from the methods section – in particular, whether both trial arms were sampled equally and how they accounted for mortality among infants with HIV, in selecting mother-infant pairs. How many time-points did each mother contribute? The reader needs to appreciate who is in the cohort over time, and how many repeated measures were included. How different were the women in both groups who provided later samples, compared to those who provided only early samples (either because they subsequently weaned or because their infant died)? Overall, how have the authors accounted for this inevitable missing data over time in the analysis? We have combined many of our responses to this comment and the previous comment in our above response given their related nature.

On the topic of repeated measures, we noted in the Methods:

“Linear mixed effects models of the form metabolite ~ HIVExposure*Visit + (1 | PID) where HIVExposure is an indicator variable for WWH versus WWoH, Visit is a categorical variable for the study visit, and PID is the unique participant identifier were used to identify differentially abundant metabolites as well as differences in KT log-ratio. Estimated marginal means were used to conduct the post hoc comparisons either averaged or stratified by study visit.”

This statistical approach is the appropriate way to account for the longitudinal nature of the data. Additionally, we also conducted stratified analyses by study visit and results were consistent. Thus, identified features are related to HIV infection and not due to differences in early versus later samples.

3. Because of the challenges for the reader in following the sampling strategy, a flow chart would be very helpful, to show the original ZEBS cohort (which presumably included women without HIV who were randomized also?), split by randomized arm, and sample collection (during the trial) and availability (following storage), and how different groups were then selected for this study. Similarly, figure 1 would benefit from some more information to show excluded samples (i.e. how the 1599 samples declined to 1426).

In addition to the changes described in the two previous comments, we have also added a new supplemental figure as **Supplementary Figure 1** that shows the excluded samples in addition to the samples shown in **Figure 1A**. This clearly displays which samples were excluded for the final analysis of 1426.

In the original ZEBS cohort, women without HIV were not randomized. Practices were identical up to 4 months.

4. The authors state that the targeted KT analyses were done at 4 months because almost all women had available breast milk samples, and there were paired plasma samples available. But only 21/32 women without HIV (66%) and 77/189 (40%) of women with HIV who had untargeted metabolomics done, also had targeted KT analysis. Why was this not done in all women, and can the authors show a supplementary table to show how similar or different the women were who had versus did not have the targeted analysis done, so readers can appreciate whether this subgroup of women were similar or different to the whole cohort.

A subset of women was selected for quantitative KT panel as follows:

- A random sample of 21 WWoH
- A random sample of 20 WWH whose infants did not experience HIV infection and who survived
- A random sample of 20 WWH whose infants did not experience HIV infection but died in the first 24 months of life
- A random sample of 20 WWH whose infants tested positive for HIV between 1 week and 1 month of life

- A random sample of 20 WWH whose infants tested positive for HIV after 1 month of life

This sample size was selected to attain 90% power to detect a Cohen's effect size of $d=0.8$.

To help clarify which samples had milk and plasma targeted KT analyses run, we have added a Venn diagram as **Supplementary Figure 8** and demographics data as **Supplementary Table 11**.

5. The linear models do not include any covariates that might be important, such as maternal age, nutritional status (e.g. MUAC) and infant sex. These are presumably all factors that might contribute to different metabolic profiles of milk?

We agree with the Reviewer that these are potentially interesting covariates; however, as none of these differed by the primary group comparison of WWH versus WWoH, it would not be appropriate to include them as confounders in the regression analyses.

We have, however, added these variables and parity to a revised PERMANOVA (**Supplementary Table 1**). Unfortunately, MUAC was not measured but we have included food unavailability and maternal BMI at 4 months postpartum as alternate measures of nutritional status in **Table 1** and **Supplementary Table 11**.

To the larger point that these may all affect the metabolic profile of milk, we wholeheartedly agree with the Reviewer and plan to study these variables in more detail in follow-up work.

6. The authors state that the Haiti cohort validates their findings in a less sick cohort. It is definitely a strength that this study had access to another cohort, albeit small, but I do not see evidence that the findings were replicated. Why is the volcano plot for the Haiti cohort not shown? Did the authors adjust for the difference in infant age between women with and without HIV, given that this was an important determinant of metabolic profiles in the larger cohort from Zambia? The only finding that is replicated with a significant adjusted p value is the elevation in cytosine. Given that this is actually the key finding across cohorts, it is not clear why the paper focuses more on KT ratios than on cytosine, and why this key (replicated) difference was not taken forwards more for further exploration? None of the adjusted P values for tryptophan or KT ratio were significant in the Haiti cohort. It is confusing in the results section to switch between adjusted and unadjusted P value results to explain the findings of the cohort.

We thank the Reviewer for these very helpful comments.

We have rerun the regression models for the Haiti cohort including infant age as a covariate and updated the Results section accordingly (**Supplementary Figure 11** and **Supplementary Table 14**). We have also added the volcano plot for the Haiti cohort as **Supplementary Figure 11B**.

Although the KT results are not statistically significant in the Haiti cohort after adjustment, the magnitude of the associations are similar. Notably, the WWH in Haiti were on ART, so we expect some differences in their metabolomic profiles. We also note that the cytosine findings were discussed, and we hypothesized that the significant elevation of cytosine in WWH from Haiti may reflect breakdown of ddhCTP:

“Cytosine was also persistently elevated in the milk of WWH. Viral infection has been associated with alterations in host cytosine metabolism¹⁰ and the production of ddhCTP¹¹. While ddhCTP is a viral chain terminator, host RNA and DNA polymerases are not affected by ddhCTP¹². We found cytosine elevations in both our primary cohort and in our validation cohort. We hypothesize that the elevated levels of cytosine in the milk of WWH reflects breakdown of ddhCTP.”

We appreciate the confusion in switching between adjusted and unadjusted p values; however, the KT ratio analysis consists of a single statistical test and therefore the unadjusted p-value is the correct comparison.

7. *Were plasma samples available from infants? It would be very intriguing to see if lower tryptophan levels in breast milk led to lower circulating plasma tryptophan and altered metabolic profiles in the infants.*

Plasma samples from infants were not included for this analysis. We agree this could be an interesting future study.

8. *Overall, the discussion section makes quite definitive claims. The results are novel, and intriguing, but the authors need to be clear that they were not replicated in a less sick cohort, and these findings may therefore simply reflect a very sick cohort of patients with advanced HIV disease, no access to ART, and 50% child mortality. A large, sub-Saharan African cohort, who are stable on ART, is really needed now to replicate these initial findings. The mechanistic explanation for the findings presented in Figure 5 feels like quite a leap, because it combines cytokine alterations (not explored in this study), with other metabolites not examined (I could not see evidence of serotonin or quinolinic acid measurements in this cohort, for example), to pull together a mechanistic explanation. The authors need to acknowledge that this study was limited in its ability to explore mechanism and Figure 5 therefore feels extremely speculative.*

We thank the Reviewer for these helpful comments and fully agree that a large, sub-Saharan African cohort of WWH stably on ART is really needed. We believe the analyses presented in the response to comment #1 show that the findings in this study are not limited to WWH with advanced HIV disease and high child mortality. We also respectfully disagree that these findings were not replicated; the estimated effects correlate very strongly between the two cohorts, and the limited sample size of the Haiti cohort is likely a major reason why statistical significance is not achieved given the high testing burden. We also clearly state in the Results that the tryptophan and kynurenine results from Haiti were concordant with ZEBS, but did not achieve statistical significance:

“Tryptophan trended lower, and kynurenine trended higher in the milk of WWH in the Haiti cohort as observed in the main cohort, although neither reached statistical significance”

We agree that the mechanistic explanation presented in **Figure 5** is speculative and have removed this Figure and all associated references from the manuscript entirely.

We have also added language in the Discussion to further acknowledge the limitations on generalizability of the present study.

The relevant section of the Discussion now reads:

“Consistent with these points, the metabolomic signature in the milk of WWH is partially observed in the validation cohort of women on antiretroviral therapy with higher CD4 counts. Replication of these findings in a large cohort of WWH and their infants in contemporary settings with universal maternal antiretroviral therapy is necessary to confirm the hypothesized mechanisms on the health outcomes of CHEU.”

9. *Overall, the discussion section needs to better acknowledge the limitations of the analysis (e.g. missing data, multiple subgroups, non-replicated findings in the smaller Haiti cohort) and make somewhat less definitive claims (ego the conclusion speculates that infants may have serotonin deficiency and altered tryptophan catabolism, but these were not even explored).*

In addition to the additions above, limitations have been expanded to address this important point:

“Finally, the present study only includes samples from WWH and WWoH with no measurements in the infants. Consequently, the effect on the infants has to be inferred as it was not directly measured.

Minor comments

1. Please add some sociodemographic data to Table 1 to contextualize the cohorts (e.g. maternal education, some marker of SES, rural versus urban) as well as nutritional status and HIV disease stage.

We have added additional sociodemographic data to **Table 1** and **Supplementary Table 11**. This table now includes maternal education, markers of SES, maternal nutritional status and maternal HIV disease stage. All study participants were recruited from two sites in a single city (Lusaka).

2. Did women with recent infections or mastitis get excluded from analysis, as these may have altered breast milk metabolites.

Women with mastitis were not excluded. Please note, no women with incident HIV infection were included. All women living with HIV were chronically infected and identified early in pregnancy.

3. I am surprised to see no authors from the countries where the cohorts were recruited.

Although we appreciate the Reviewer’s diligence in ensuring appropriate credit for this work, this study is based on analysis of samples obtained from a biorepository. The primary manuscript was published in 2008 (Kuhn L et al. NEJM 2008)¹³ with appropriate authorship for contributors at the study sites.

Reviewer #3 (Remarks to the Author):

I would like to thank the authors for submitting their manuscript and for addressing an important topic. The manuscript investigates a critical knowledge gap, in understanding the human milk metabolome of women with HIV (WWH) compared to women without HIV (WWoH). The study profiled the milk metabolome of WWH and WWoH and identified decreased and increased levels of tryptophan and kynurenine in WWH compared to WWoH as well as a high tryptophan: kynurenine ratio in WWH.

However, after careful consideration, I have several major concerns that, in my view, significantly limit the suitability of the manuscript for publication in its current form.

1. Title: *The study primarily focused on the milk metabolome. There is no analysis or results to show how the findings relate to the health of children born to women with HIV.*

We have changed the title to “Altered milk tryptophan and tryptophan metabolites in women with HIV”.

2. Abstract: *“Our findings provide intriguing evidence that decreases in tryptophan availability and perturbations in tryptophan catabolism in children born to WWH may contribute to their increased morbidity and mortality”. While this is a promising hypothesis, the study did not test this and giving this statement feels a bit of an overreach.*

We thank the Reviewer for this feedback and agree that such a statement is premature for this study. We have edited this part of the Abstract to now read:

“Our findings suggest that decreases in milk tryptophan availability and perturbations in milk tryptophan catabolism in WWH may contribute to the increased morbidity and mortality of children born to WWH.”

3. Introduction:

a. *The authors indicated that they used a healthier cohort of WWH from Haiti for validation. “Lastly, we confirm this milk metabolomic signature of HIV in a second, healthier cohort of WWH on antiretroviral therapy”. How this healthy cohort was defined is not clear.*

The second, healthier cohort referenced here is the Haiti cohort (bottom portion of Table 1). Plasma viral loads and maternal baseline CD4 counts highlight the relative disparity in health between the two cohorts. We have clarified this in the Introduction as follows:

“Lastly, we confirm this milk metabolomic signature of HIV in a second, healthier cohort of WWH on antiretroviral therapy from Haiti (see study overview, Figure 1)”

b. *The authors have not indicated in the introduction whether the tryptophan level in milk changes over time during the lactation period. This would enrich the readers to understand tryptophan fluctuations during infancy given they conducted a longitudinal study through 18 months of infancy. We thank the Reviewer for this helpful suggestion as we are also quite interested in the longitudinal profiles of the various metabolites identified in this study, including tryptophan.*

However, to the best of our knowledge, there are only a few studies describing tryptophan fluctuations in milk and over infancy. Breast milk contains both free and protein-bound tryptophan and levels of both forms in the milk decline quickly as the milk changes from colostrum to transitional milk measured days 3, 4, 5, 10, and 30 (see Zanardo V et al. *Biol Neonate* 1989)¹⁴(Kamimura S et al. *Acta Medica Okayama* 1991)¹⁵. Lower levels of free tryptophan are seen in the milk of preterm infants with increased inflammatory markers measured at 7 and 14 days postpartum (O’Rourke et al. *J Nutr Sci* 2018)¹⁶. One study of plasma tryptophan levels measured at 4 weeks and 12 weeks of life showed no variation in tryptophan levels (Fazzolari-Nesci et al. *JPGN* 1992)¹⁷.

We have added the following sentence to the introduction:

“There are very few reports describing levels of tryptophan in milk, it appears that tryptophan, free and bound, are higher in colostrum and then decline with time.”

4. Methods:

a. *The current study analysed 228 WWH and 36 WWoH selected from a clinical trial. The authors did not provide the number of participants enrolled in the original study. It’s not known what fraction of the participants from the clinical trial were analysed in the current study. Is the proportion of participants analysed in this study reflective of the original trial design?*

We thank the Reviewer for this helpful question. Briefly, we selected all available WWoH (38), all available WWH whose infants died while remaining uninfected (78), and all available WWH whose children acquired HIV via intrapartum (53) or postnatal (81) infection. We also included a random sample of WWH whose children survived the study course while remaining uninfected (76 selected out of 622 total). This was described in the Methods under the “Breast milk sample selection for the global metabolomics panel:” sub-section.

We agree with the Reviewer that the existing description was unclear and made it difficult to ascertain the actual selection of participants and samples. Therefore, we have amended the Methods section to use a list format of the exact groups selected for this study. The relevant section now reads:

“Longitudinal breast milk samples spanning the first 24 months of lactation from women living without (N=38) and with HIV (N=288) were selected on the basis of sample availability. Samples were selected from all 38 women living without HIV (WWoH) included in the trial, 35 of their infants

survived (279 samples) and 3 infants died (9 samples). A stratified sample of WWH were selected for inclusion by known infant outcomes for this metabolomics study. Specifically, the samples selected for study inclusion were as follows:

- All WWoH (N=38; 288 samples)
- A random sample of 76 (out of 622) WWH with infants who did not experience HIV infection and who survived (N=76; 396 samples)
- All WWH whose infants did not experience HIV infection but died in the first 24 months of life (N=78; 193 samples)
- All WWH whose infants tested positive for HIV between 1 week and 1 month of life (N=53; 221 samples) or after 1 month of life (N=81; 401 samples)."

Our revisions substantially improve the clarity of how participants were selected from the parent ZEBS trial.

b. The only selection criterion indicated was that the WWH had not ceased breastfeeding at 4.5 months, however, it is unclear whether these mothers continued to breastfeed until the end of the observation period which is 18 months. This is critical as breastfeeding itself changes milk output and its composition.

In the original ZEBS study, WWH were randomized to either abruptly stop breastfeeding at 4 months or to continue breastfeeding for their preferred duration. WWoH were not randomized and were told to breastfeed for their preferred duration. We agree with the Reviewer that it is important to control for the effect of breastfeeding itself on the milk composition, which is why we excluded all milk samples that were collected post-weaning for the final analysis of n=1426 samples. This is now more clearly shown in **Supplementary Figure 1**.

c. The data was collected in the pre-antiretroviral (ART) era where ART coverage had not widely spread, however, the authors indicated that most of the women in the current analysis received a single dose of nevirapine for prevention of vertical transmission per standard of care. Authors could provide the percentages of WWH who received nevirapine because nevirapine usage alone can alter the milk metabolome.

We have added the percentages receiving nevirapine (96.2%) to **Table 1** and **Supplementary Table 11**. The Reviewer is correct that single dose nevirapine usage at the time of delivery could alter the milk metabolome. However, many studies of nevirapine concentrations in milk and other compartments (see

Kunz A et al. J Antimicrob Chemother 2009)¹⁸ (Musoke P et al. AIDS 1999)¹⁹ (LactMed®)²⁰, among others, indicate that its effect would likely only been seen at the 1 week and possibly 1 month visits. We therefore think it is unlikely that the differences observed in WWH versus WWoH across the first 18 months of infancy are strongly influenced by this single dose of nevirapine.

d. Among the criteria for exclusion and inclusion of samples was the weaning period i.e., samples collected post-weaning period were excluded (n = 99). However, the authors did not define the weaning period in this population. Additionally, the readers would benefit if the authors could provide a concise consort on the exclusion and inclusion of participants for different analyses done in the current study.

Women living with HIV were randomized to either breastfeeding for their preferred duration or to abrupt cessation of breastfeeding at 4 months. Women without HIV were not randomized and breastfed for their preferred duration. We have updated the relevant part of the Methods section, and it now reads:

“To account for this study intervention, only milk samples from women who reported they were actively still breastfeeding were included in the analysis.”

We have also added another study schematic as **Supplementary Figure 1** which shows exactly what samples were excluded as well as the reason for exclusion and the timing of the sample collection. We hope that this helps to clarify which samples and participants were used for the main analysis of WWH versus WWoH.

We have also added **Supplementary Figure 8 and Supplementary Table 11** to clarify which samples were included in the quantitative KT panel analyses of milk and plasma.

5. Results:

a. Table 1 presents the baseline characteristics of the study participants including continuous variables such as maternal age, CD4 and CD8 counts, and infant birth weight. It is confusing whether these continuous variables were summarised as median (interquartile ranges [IQR]) or mean (standard deviations [SD]). Currently, the authors presented them as median (SD) which is confusing. This is also seen in Supplementary Tables and Methods. Please clarify this. Median is a non-parametric measure, whereas SD is a parametric measure. They do not naturally combine. We have changed the summary statistics for all continuous variables to be mean (standard deviation).

b. Table 1: what is the unit of measure for parity?

The unit of measure for parity is pregnancies resulting in a live birth.

c. “WWH had lower CD4 T cell counts (315 vs. 840 cells/mm³) and higher CD8 T cell counts (759 vs 563 cells/mm³) than WWoH ($p < 0.001$ and $p = 0.001$, respectively)”. These values in the text don't match the values indicated in Table 1 under maternal CD4 and CD8 count.

We thank the Reviewer for catching this inconsistency. The values in the text were the mean values, and those in Table 1 were the medians. We have corrected this by changing all of the values in Table 1 (and other tables) to show mean (SD).

d. Table 1: for consistency, the authors could indicate the p-value for the difference in gestational age between WWoH and WWH. For the rest of the variables in the table, the authors computed their p-values

We thank the Reviewer for catching this omission. The p-value has been added to Table 1.

e. Table 1: See the below excerpt of Table 1. What does asterisk (*) mean? You have indicated that the p-value is NA. Should this be the case?

Value WWoH WWH p-value

Child death (%) 3 (7.9) 145 0.3)* NA

We apologize for this oversight. The description of the asterisk was mistakenly dropped in revisions.

We have added this back as a footnote in Table 1:

“* Based on sampling strategy”

f. The authors indicated that broadly the milk metabolic profiles were strongly influenced by study visits, HIV infection, maternal CD4 count, and infant sex. “Overall milk metabolomic profiles were strongly influenced by study visit (4.0% variance, $p < 0.001$), and by HIV infection (0.2% variance, $p < 0.001$), maternal CD4 count (0.15% variance, $p < 0.001$), and infant sex (0.07% variance, $p = 0.009$) (Supplementary Fig. 1)”. The cited Supplementary Fig. 1 by the authors shows PC1 and PC2 of metabolite clustering coloured by maternal HIV status (WWH and WWoH). From this figure, there is no clear separation of metabolites by maternal HIV status as the ellipses showing 95% confidence areas of WWH and WWoH greatly overlap. Further, there is no information about study visits, maternal CD4 count, and infant sex on metabolomic variance in this Figure as cited by the authors. The reported variances and their corresponding significance levels are not provided by the authors in Supplementary Fig. 1

We have added the PERMANOVA results as the new **Supplementary Table 1**.

g. Authors reported that linear mixed effect regression identified 173 metabolites significantly altered

in WWH compared to WWoH across all the study visits and cited Supplementary Table 1. However, from Supplementary Table 1 at an adjusted p-value of <0.05 as indicated in the methods, 104 metabolites and not 173 metabolites were significant. Could the authors clarify this discrepancy? We apologize for the oversight as the 173 metabolites was based on an adjusted p-value of <0.1 and we did not update the number accordingly in the text. We have now clarified the text to read:

“Linear mixed effects regression identified 103 metabolites with significantly altered levels in WWH versus WWoH when all study visits were considered in aggregate (**Supplementary Table 2**).”

h. “Milk levels of tryptophan were 6.8% and 8.3% of the plasma levels for WWoH and WWH, respectively; this proportion did not differ significantly by maternal HIV infection ($p=0.14$, Supplementary Fig. 8A and Supplementary Table 8). A similar trend was observed for kynurenine with breast milk levels at 32.4% and 36.6% of the corresponding plasma levels for WWoH and WWH, respectively ($p=0.42$, Supplementary Fig. 8B and Supplementary Table 8). Here, the authors are reporting results of the Milk/Plasma ratio for tryptophan or kynurenine, which according to the authors’ definition investigated the transfer of tryptophan or kynurenine from the bloodstream to milk. This computation gives a “ratio” which may not directly translate to proportions. The authors reported proportions while they cited Supplementary Figure 8A which indicates the boxplots of concentration of tryptophan and kynurenine in plasma and milk samples stratified by maternal HIV status. These are median values and not proportions. In Supplementary Table 8 the author reported tryptophan or kynurenine availability computed by Milk / Plasma Ratio as shown below. Could the authors explain whether this ratio directly translates to proportion and its interpretation?

Table 8. Quant KT

WWH WWoH p-value

Tryptophan, median (sd)

Milk 0.318 (0.177) 0.620 (0.222) 0.00082

Plasma 5.110 (2.357) 8.380 (2.298) 1.29E-07

Milk / Plasma Ratio 0.083 (0.038) 0.068 (0.025) 0.14

Kynurenine, median (sd)

Milk 0.147 (0.117) 0.109 (0.084) 0.028

Plasma 0.453 (0.158) 0.362 (0.115) 0.00065

Milk / Plasma Ratio 0.366 (0.220) 0.324 (0.199) 0.42

We are using the Milk/Plasma ratio for tryptophan and kynurenine as a measure of the proportion that is transferred from the bloodstream to milk. These were shown on the right-hand panels of Supplementary Figure 8A and 8B whereas the left-hand panels show the boxplots of concentrations as noted by the Reviewer. We appreciate that this was a confusing way to organize the figure, and have reorganized it so that all of the concentration data are now in part A, and all of the ratio data are now in part B.

Unrelated to this comment, we have added two supplemental figures, so what was previously Supplementary Figure 8 is now **Supplementary Figure 10**. We have reworded the corresponding part of the Results section and the legend for the new Supplementary Fig 10 accordingly.

We have also added four additional supplemental tables, so what was previously Supplementary Table 8 is now **Supplementary Table 12**. The values shown in the new Supplementary Table 12 can be directly matched to the new Supplementary Fig 10. The Milk and Plasma rows for tryptophan and kynurenine correspond to the values shown in Supplementary Figure 10A. The Milk / Plasma Ratio rows for tryptophan and kynurenine correspond to the values shown in what is now Supplementary Figure 10B.

We have also updated the values in Supplementary Figure 10 and Supplementary Table 12 to reflect means instead of medians.

i. The random forest results do not contribute anything to the paper.

We respectfully disagree with the Reviewer on this point. The random forests results provide confirmatory evidence for the linear regression analyses identifying key metabolites that differentiate WWH and WWoH. Furthermore, as random forests is a nonlinear approach, it adds value in showing that both linear and nonlinear methods identify the same set of key metabolites that are perturbed in the milk of WWH.

j. The study is limited to the investigation of free tryptophan. However, essential amino acids could also be incorporated into albumin and other proteins in the breastmilk, which will be digested by the infant. Arguing that the lower free tryptophan levels in this study could equate to having poor health outcomes to the child is not supported by data.

The reviewer correctly notes that metabolomics, both the global panel and the quantitative KT measurements, measure free tryptophan only. Breast milk contains both free and protein-bound tryptophan and levels of both forms in the milk decline quickly as the milk changes from colostrum to transitional milk (see Zanardo V. Biol Neonate 1989). However, it is only free tryptophan that is the form that crosses the blood-brain barrier for the production of brain serotonin. While we acknowledge that measures in breast milk are only an indirect marker of what is occurring in the infant, we note that in the early months of life tryptophan in breast milk is the only source for infants. Nevertheless, we have added additional caveats to the discussion.

“These findings raise the question of whether replacement of tryptophan in the diet of children of WWH, if low tryptophan is found in investigations of CHEU, will lead to resolution of the immune, cognitive and growth perturbations experienced by this population.”

and

“Finally, the present study only includes samples from WWH and WWoH with no direct measurements in the infants. Consequently, the effect on the infants has to be inferred as it was not directly measured. Replication of these findings in a large cohort of WWH and their infants in contemporary settings with universal maternal antiretroviral therapy is necessary to confirm the hypothesized mechanisms on the health outcomes of CHEU.”

It is not known if the low levels of tryptophan in the milk are still sufficient for the infants, how well the tryptophan is absorbed by the infant in the setting of exposure to chronic viral inflammation, or if the systemic inflammation which has been characterized in CHEU infants leads to alterations in tryptophan catabolism with an increase in neurotoxic metabolites. All of these possibilities deserve careful study.

Reviewer #4 (Remarks to the Author):

1. Huengsberg M, Winer JB, Gompels M, Round R, Ross J, Shahmanesh M. Serum kynurenine-to-tryptophan ratio increases with progressive disease in HIV-infected patients. *Clin Chem*. Apr 1998;44(4):858-62.

2. Bipath P, Levay PF, Viljoen M. The kynurenine pathway activities in a sub-Saharan HIV/AIDS population. *BMC Infect Dis*. Aug 19 2015;15:346. doi:10.1186/s12879-015-1087-5
3. Favre D, Mold J, Hunt PW, et al. Tryptophan catabolism by indoleamine 2,3-dioxygenase 1 alters the balance of TH17 to regulatory T cells in HIV disease. *Sci Transl Med*. May 19 2010;2(32):32ra36. doi:10.1126/scitranslmed.3000632
4. Mehraj V, Routy JP. Tryptophan Catabolism in Chronic Viral Infections: Handling Uninvited Guests. *Int J Tryptophan Res*. 2015;8:41-8. doi:10.4137/IJTR.S26862
5. Silvano A, Seravalli V, Strambi N, et al. Tryptophan degradation enzymes expression in the placenta and the Kynurenine/Tryptophan ratio in maternal plasma after elective cesarean section. *J Reprod Immunol*. Mar 2023;156:103823. doi:10.1016/j.jri.2023.103823
6. Collins JM, Siddiq A, Jones DP, et al. Tryptophan catabolism reflects disease activity in human tuberculosis. *JCI Insight*. May 21 2020;5(10)doi:10.1172/jci.insight.137131
7. Gazi MA, Das S, Siddique MA, et al. Plasma Kynurenine to Tryptophan Ratio Is Negatively Associated with Linear Growth of Children Living in a Slum of Bangladesh: Results from a Community-Based Intervention Study. *Am J Trop Med Hyg*. Nov 23 2020;104(2):766-773. doi:10.4269/ajtmh.20-0049
8. Johnson KE, Hernandez-Alvarado N, Blackstad M, et al. Human cytomegalovirus in breast milk is associated with milk composition and the infant gut microbiome and growth. *Nat Commun*. Jul 23 2024;15(1):6216. doi:10.1038/s41467-024-50282-4
9. Thea DM, Aldrovandi G, Kankasa C, et al. Post-weaning breast milk HIV-1 viral load, blood prolactin levels and breast milk volume. *AIDS*. Jul 13 2006;20(11):1539-47. doi:10.1097/01.aids.0000237370.49241.dc
10. Blasco H, Bessy C, Plantier L, et al. The specific metabolome profiling of patients infected by SARS-COV-2 supports the key role of tryptophan-nicotinamide pathway and cytosine metabolism. *Sci Rep*. Oct 8 2020;10(1):16824. doi:10.1038/s41598-020-73966-5
11. Danchin A, Marliere P. Cytosine drives evolution of SARS-CoV-2. *Environ Microbiol*. Jun 2020;22(6):1977-1985. doi:10.1111/1462-2920.15025
12. Gizzi AS, Grove TL, Arnold JJ, et al. A naturally occurring antiviral ribonucleotide encoded by the human genome. *Nature*. Jun 2018;558(7711):610-614. doi:10.1038/s41586-018-0238-4
13. Kuhn L, Aldrovandi GM, Sinkala M, et al. Effects of early, abrupt weaning on HIV-free survival of children in Zambia. *N Engl J Med*. Jul 10 2008;359(2):130-41. doi:10.1056/NEJMoa073788
14. Zanardo V, Bacolla G, Biasiolo M, Allegri G. Free and bound tryptophan in human milk during early lactation. *Biol Neonate*. 1989;56(1):57-9. doi:10.1159/000242987
15. Kamimura S, Eguchi K, Sekiba K. Tryptophan and its metabolite concentrations in human plasma and breast milk during the perinatal period. *Acta Med Okayama*. Apr 1991;45(2):101-6. doi:10.18926/AMO/32183
16. O'Rourke L, Clarke G, Nolan A, et al. Tryptophan metabolic profile in term and preterm breast milk: implications for health. *J Nutr Sci*. 2018;7:e13. doi:10.1017/jns.2017.69
17. Fazzolari-Nesci A, Domianello D, Sotera V, Raiha NC. Tryptophan fortification of adapted formula increases plasma tryptophan concentrations to levels not different from those found in breast-fed infants. *J Pediatr Gastroenterol Nutr*. May 1992;14(4):456-9. doi:10.1097/00005176-199205000-00014
18. Kunz A, Frank M, Mugenyi K, et al. Persistence of nevirapine in breast milk and plasma of mothers and their children after single-dose administration. *J Antimicrob Chemother*. Jan 2009;63(1):170-7. doi:10.1093/jac/dkn441
19. Musoke P, Guay LA, Bagenda D, et al. A phase I/II study of the safety and pharmacokinetics of nevirapine in HIV-1-infected pregnant Ugandan women and their neonates (HIVNET 006). *AIDS*. Mar 11 1999;13(4):479-86. doi:10.1097/00002030-199903110-00006
20. Drugs and Lactation Database (LactMed) [Internet]. Bethesda (MD): National Institute of Child Health and Human Development; 2006-. Nevirapine. [Updated 2024 Oct 15]. Available from: <https://www.ncbi.nlm.nih.gov/books/NBK501535/>

We would like to thank the reviewers' for their careful consideration of the manuscript and thoughtful critiques that have significantly improved the manuscript.

Reviewer #1 (Remarks to the Author):

In this revised manuscript, Tobin et al. have fully addressed all previously suggested changes, substantially improving the clarity and emphasis on the study's longitudinal design and validation approach. The authors now more effectively highlight the depth and nuance of their rich dataset, particularly in relation to (a) the longitudinal validation of tryptophan dysregulation in women with HIV, (b) the annotation of 3'-deoxy-3'4'-didehydro-cytidine (ddhC), and (c) the cross-cohort biomarker validation using samples from Haiti. These refinements further underscore the significance of their findings and the strength of the study's design. The manuscript now presents a clear, compelling, and rigorously supported narrative on the impact of HIV on maternal and infant health via breast milk. I recommend acceptance of this manuscript for publication in Nature Communications.

Thank you

Reviewer #2 (Remarks to the Author):

The authors have responded to my queries adequately

Thank you

Reviewer #3 (Remarks to the Author):

I appreciate the efforts made by the authors in revising the manuscript based on the reviewers' suggestions.

However, I still believe that the abstract (albeit unintentionally) overrepresents the results. The statement "Our findings suggest that decreases in milk 59 tryptophan availability and perturbations in milk tryptophan catabolism in WWH may contribute to the increased morbidity and mortality of children born to WWH." appears to have been revised, but actually still means the same. The study does not suggest anything about the morbidity or mortality of children, neither was it measured directly or indirectly. I suggest the following text: "Our findings demonstrate that milk tryptophan content and availability decreases among WWH, which may indicate perturbations in milk tryptophan catabolism. The link between this perturbation and the increased morbidity and mortality of children born to WWH merits further investigation."

We have made the suggested changes to the Abstract. The relevant section now reads:

“Finally, we validate these signatures of maternal HIV infection in an independent cohort of healthier WWH. Taken together, our findings demonstrate that milk tryptophan content and availability decrease among WWH, which may indicate perturbations in milk tryptophan catabolism. The link between this perturbation and the increased morbidity and mortality of children born to WWH merits further investigation.”

My opinion about the random forest results still stand. I do not believe that it adds anything to the manuscript - the results are just as strong without it. In the answer of the authors to the comment, they said: "We respectfully disagree with the Reviewer on this point. The random forests results provide confirmatory evidence for the linear regression analyses identifying key metabolites that differentiate WWH and WWHoH. Furthermore, as random forests is a nonlinear approach, it adds value in showing that both linear and nonlinear methods identify the same set of key metabolites that are perturbed in the milk of WWH".

Random forest does not provide the best confirmatory evidence for the association between tryptophan and WWH. The confirmatory evidence you need is to see the same association in a different cohort, which you have. Furthermore, the authors argue that RF is a non-linear approach, showing both linear and non-linear methods agree with the results. This is not entirely correct. Random forest is ABLE TO hand/uncover non-linear associations, but the actual model it fits could also be linear. There is no telling whether the model fit a linear or non-linear association in your data.

The results are just as good without the random forest model. I know it is fancy, but not needed. I refer to the decision of the editor about this issue.

We defer to the Editor on this comment. We prefer that the random forest modeling remain in the manuscript. We believe that it strengthens the findings to show the selection of similar features in more than one model. The figures, supplementary figures 5 and 6, combining features selected by the random forest models with the estimates from the emmeans model serve as a strong visual demonstration of the robust signature of HIV in milk over the 18 months of the study. Additionally, supplementary figure 6 showing the sub-analyses of WWH whose children remained uninfected throughout the course of the study (CHEU; Supplementary Fig. 6a) and those who survived for the duration of the study (Supplementary Fig. 6b) confirmed that the signature seen in the milk was independent of child outcome. This is most easily visualized by comparing supplementary figures 5 and 6.

On my remark about having only free tryptophan analysed in the study, the authors responded: " it is only free tryptophan that is the form that crosses the blood-brain barrier for the production of brain serotonin". I agree. BUT, free tryptophan could also be

formed from the bound tryptophan during gastrointestinal digestion. This should be considered as well in the manuscript.

This important consideration has been added to the sentence below (in red).

The authors further answered: "It is not known if the low levels of tryptophan in the milk are still sufficient for the infants, how well the tryptophan is absorbed by the infant in the setting of exposure to chronic viral inflammation, or if the systemic inflammation which has been characterized in CHEU infants leads to alterations in tryptophan catabolism with an increase in neurotoxic metabolites. All of these possibilities deserve careful study." I propose to also include this sentence in the manuscript.

We have added *these sentences* to the Discussion. The relevant section now reads:

*"These findings raise the question of whether replacement of tryptophan in the diet of children of WWH, if low tryptophan is found in investigations of CHEU, will lead to resolution of the immune, cognitive and growth perturbations experienced by this population. It is not known if the low milk tryptophan levels are sufficient, how well the tryptophan is absorbed, or **the proportion of bound tryptophan in the milk**. Systemic inflammation has been described in CHEU infants which may alter tryptophan catabolism leading to increases in neurotoxic metabolites. These possibilities deserve careful study. Breastfeeding remains the recommended form of infant feeding because health benefits outweigh the risk of HIV transmission."*

Reviewer #4 (Remarks to the Author):

Reviewer #4 (Remarks on code availability):

The codes are available, but access to the data requires a private link which I do not have. Therefore, I have not ascertained if the codes and results are reproducible.

We have made all associated code and data repositories publicly available.

ROUND 1 REVIEWER 1 ATTACHMENT:

Review of

Altered milk tryptophan and tryptophan metabolites and health of children born to women with HIV

In this manuscript, Tobin et al. present an exceptional longitudinal study of breast milk samples for women with HIV (WWH) and women without HIV (WWoH) through an initial investigation of a cohort in Zambia that was validated by samples collected from a cohort in Haiti. Through this work, the authors were able to make several significant strides in elucidating the effects of HIV on infants through breast milk. These include the (a) longitudinal validation of tryptophan dysregulation in WWH, (b) the annotation of 3'-deoxy-3'4'-didehydro-cytidine (ddhC), and (c) the validation of biomarkers in a separate Haitian cohort. Collectively, the text and figures together tell a clear and compelling story of HIV's effects on maternal and infant health. However, the unique insights from the longitudinal sampling and validation study currently feel underemphasized. This reviewer recommends **major revisions** to better highlight the depth and nuance of this rich dataset. With these revisions, the manuscript would be a strong contribution to Nature Communications and its broad readership.

Revisions and Comments:

1. Figure images appear pixelated in the rendered pdf and should be reviewed before publication.
2. Can the authors clarify why the 1- and 4-month time points were highlighted in the Zambia cohort? Are these associated with specific biological or physiological events? Other time points appear to show more pronounced molecular changes, so the focus on these two should be justified in the text.
3. In Figure 2, 3 and several supplemental figures, the authors show normalized abundances of several molecules across time points with black and red stars on each plot. The meaning of these stars is unclear and should be specified in the text.
4. It is recommended that Figure 1E be amended to include the timeline of sample collection for the Haiti validation cohort. Additionally, a justification for comparison to the 4-month timepoint from the Zambia cohort could be provided within the methods text.
5. The authors mention 9,10-DiHOME and 12,13-DiHOME metabolites were significantly elevated in WWH at 1 month. These molecules are, however, isomers and indistinguishable with MS. It should be clarified in text how these molecules are being annotated as the respective isomers observed?
6. Supplemental Figures 6 and 7, which appear to go alongside the KT ratio results (Figure 3) of the main text show continued significance of tryptophan (assuming stars indicate significance). Alternatively, kynurenine does not appear significant at any time point. The authors should justify in text the continued significance of the KT ratio if kynurenine is not significant.

7. The authors note that this study uniquely provides longitudinal insights into metabolic changes in breast milk from WWH and WWoH, which is certainly a strength. However, the longitudinal aspects feel underemphasized in the main text. In contrast, the supplemental figures reveal a richer temporal dimension, with several compound groups clearly identified at specific time points. These observations could be further emphasized by the authors to highlight the longitudinal insights from this work.